# Repeated multi-domain cognitive training prevents cognitive decline, anxiety and amyloid pathology found in a mouse model of Alzheimer disease

Jogender Mehla[1,2], Scott H. Deibel[1,3], Hadil Karem[1], Nancy S. Hong[1], Shakhawat R. Hossain[1], Sean G. Lacoursiere[1], Robert J. Sutherland[1], Majid H. Mohajerani [1,4✉] & Robert J. McDonald [1,4✉]

Education, occupation, and an active lifestyle, comprising enhanced social, physical, and mental components are associated with improved cognitive functions in aged people and may delay the progression of various neurodegenerative diseases including Alzheimer's disease. To investigate this protective effect, 3-month-old APP$^{NL-G-F/NL-G-F}$ mice were exposed to repeated single- or multi-domain cognitive training. Cognitive training was given at the age of 3, 6, & 9 months. Single-domain cognitive training was limited to a spatial navigation task. Multi-domain cognitive training consisted of a spatial navigation task, object recognition, and fear conditioning. At the age of 12 months, behavioral tests were completed for all groups. Then, mice were sacrificed, and their brains were assessed for pathology. APP$^{NL-G-F/NL-G-F}$ mice given multi-domain cognitive training compared to APP$^{NL-G-F/NL-G-F}$ control group showed an improvement in cognitive functions, reductions in amyloid load and microgliosis, and a preservation of cholinergic function. Additionally, multi-domain cognitive training improved anxiety in APP$^{NL-G-F/NL-G-F}$ mice as evidenced by measuring thigmotaxis behavior in the Morris water maze. There were mild reductions in microgliosis in the brain of APP$^{NL-G-F/NL-G-F}$ mice with single-domain cognitive training. These findings provide causal evidence for the potential of certain forms of cognitive training to mitigate the cognitive deficits in Alzheimer disease.

[1] Canadian Centre for Behavioural Neuroscience, University of Lethbridge, Lethbridge, AB, Canada. [2] Present address: Department of Neurological Surgery, Washington University School of Medicine, St. Louis, MO 63110, USA. [3] Present address: Department of Psychology, University of New Brunswick, POB 4400 Fredericton, NB E3B 3A1, Canada. [4] These authors contributed equally: Majid H. Mohajerani, Robert J. McDonald. ✉email: mohajerani@uleth.ca; r.mcdonald@uleth.ca

Alzheimer's disease (AD), the most common form of dementia, is a progressive neurodegenerative disease affecting elderly populations worldwide. AD is characterized by extracellular amyloid-beta (Aβ) plaque deposition and intracellular formation of neurofibrillary tangles (NFT), synaptic loss, and severe cognitive decline[1,2]. Several non-genetic risk factors such as diabetes mellitus, obesity, hypertension, brain injuries, depression, or physical inactivity are related to one-third of AD cases worldwide[3–6]. Available pharmacological therapies provide only mild and temporary symptomatic relief[7]. Several epidemiological and clinical studies revealed that education, occupation, and physical activity can improve cognitive ability in healthy older people and provide protection against the development and progression of AD[5,8–11].

Several human experimental studies have reported that cognitive and physical activities prevent AD disease progression and improve cognitive functions by reducing cerebral Aβ plaques and amyloid angiopathy[12–15]. Cognitive stimulation seems to improve overall cognitive performance, especially in persons with mild-to-moderate dementia, which persisted in a 3-month follow-up. These effects are above the beneficial effects of any medication[16,17]. Some evidence also suggests that increasing mental activity in aging people shows a favorable and novel non-pharmacological approach to prevent/or delay age-related cognitive dysfunctions and decrease the number of cases of dementia[18]. Similar effects of cognitive training have been shown in rodents. Mice exposed to social, physical, and cognitive training showed protective effects against cognitive impairment, decreased brain Aβ burden, and enhanced hippocampal synaptic immunoreactivity[14,19]. Interestingly, repeated training in the Morris water task (MWM), a spatial navigation task, also induces learning improvements for newly acquired platform locations and reduces tau and Aβ pathology in 3xTg-AD animals[14]. Martinez-Coria and colleagues also reported that recurrent training in MWM ameliorated both spatial and non-spatial forms of memory in 3xTg-AD mice[19].

In the present study, we sought to clarify some issues concerning the effects of cognitive training on AD memory impairments and pathology. First, many of the human studies are epidemiological and, as such, are correlational in nature. Second of the human experimental studies in which causation can be inferred many of them manipulate multiple parameters like cognitive training, physical activity, and brain pharmacology and, thus, it is difficult to ascertain which of these factors is driving the improved function and reduced pathology. A similar pattern emerges in animal literature. Third, many of these studies investigate the effects of these lifestyle factors in aging and not AD patients specifically. These different forms of age-related cognitive decline represent different brain conditions. Fourth, studies on humans are usually confounded with a sizeable variation of socioeconomic status among participants[20]. Fifth, the animal models of AD used in the literature vary and many of these models are not ideal for various reasons, including overexpression of protein fragments that are not found in human AD[21,22]. Sixth, there is no study where the effect of multi-domain cognitive training on pathologies and cognitive deficits in experimental models of AD has been reported. Finally, many of the studies use single-domain training paradigms, which may or may not be sufficient to reduce severe memory and brain pathologies associated with AD.

Our goal in designing these experiments was to select a task that would preferentially and repeatedly activate one learning and memory network (single-domain training) and compare the effects of this brain activation during disease progression to subjects exposed to multiple tasks that would preferentially and repeatedly activate three different learning and memory networks (multi-domain training). There is a large body of evidence in mice, rats, monkeys, and humans that there are multiple learning and memory networks that are centered on key brain regions. The hippocampus is a central player in a learning and memory network involved in episodic memory. The amygdala is a key player in a network important for emotional learning and memory functions. Finally, the perirhinal cortex is a key component of a learning and memory network important for object memory (see review by McDonald et al.[23]). The hypothesis is that repeatedly activating multiple networks would be more beneficial than a single network.

As noted above, first-generation transgenic mouse models of AD overexpressed APP and APP fragments which may be responsible for artificial phenotypes in mice[21,24]. Saito et al. developed a single APP knock-in (APP-KI) mouse model for AD to overcome problems with APP overexpression[21]. APP$^{NL-G-F/NL-G-F}$ mice showed typical Aβ pathology, neuroinflammation, cholinergic dysfunction, and neurobehavioral impairment[21,25,26]. This mouse model is being used more commonly than other APP-KI lines as it develops Aβ pathology faster[21] and can be used to study various downstream mechanisms such as neuroinflammation[27,28], pericyte signaling[29], oxidative stress[30], tau propagation[31], and spatial memory impairment[25,26,32]. This is the mouse model of AD that we have selected for the present study.

With these considerations in mind, the present experiments used a new-generation knock-in mouse model of AD to assess the potential positive impacts of repeated single-domain cognitive training (ST) versus repeated multi-domain cognitive training (MT). An experimental design for the study is shown in Fig. 1. APP$^{NL-G-F/NL-G-F}$ and controls were exposed to ST or MT at 3, 6, & 9 months of age. ST was limited to a spatial navigation task. MT consisted of a spatial navigation task, object recognition, and fear conditioning. At the age of 12 months, cognitive functions were assessed for all groups. After completion of behavioral testing, mice were sacrificed, and their brains were assessed for multiple pathologies associated with AD including amyloid plaques, microglial activation, and cholinergic function. Our working hypothesis was that MT would be the most effective at preventing AD-related cognitive impairments and associated brain pathologies.

## Results

### Effect of cognitive training on various learning and memory functions of APP$^{NL-G-F/NL-G-F}$ mice

*Morris water maze.* A mixed model ANOVA with day as the repeated measures factor, and group as the between subjects'

## Single-domain cognitive training (ST)

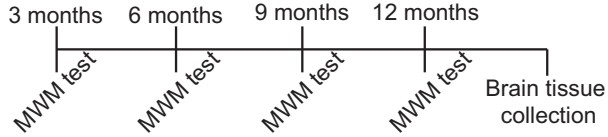

## Multi-domain cognitive training (MT)

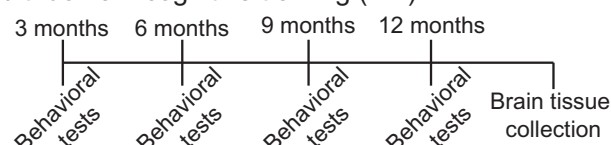

**Fig. 1 Experimental design for the study.** The study was conducted for 12 months. For multi-domain cognitive training (MT), we performed the spatial version of the Morris water task (MWM), novel object recognition (NOR) and fear conditioning (tone and context) tests. We used only MWM for single-domain cognitive training (ST).

factor was used to analyze acquisition. Latencies significantly decreased across training days, indicating that learning occurred ($F_{(7, 217)} = 14.775$, $P < 0.001$). The swim path for all experimental groups is shown in Fig. 2a. Performance differed between the groups ($F_{(3, 31)} = 16.499$, $P < 0.001$), but Sidak's multiple comparisons confirmed that only controls performed significantly better than APP-NT animals ($P < 0.001$). These data suggest that except for the APP-NT, all groups learned the MWM similarly (Fig. 2b).

Thigmotaxis behavior of mice during acquisition phase was also analyzed (Fig. 2c). We also found that training during acquisition phase caused a statistically significant ($F_{(7,217)} = 12.262$, $P < 0.000$) reduction in thigmotaxis behavior of experimental mice but this effect appears to come mostly from the APP-NT group and a comparison ($F_{(3,31)} = 8.168$, $P < 0.001$) of individual training days confirmed that APP-NT group showed significantly more thigmotaxia than other experimental groups and post-hoc comparisons confirmed this effect on some days ($P < 0.05$) (Fig. 2d). Interestingly, no significant difference was found in thigmotaxia between all experimental groups on day 8 of the acquisition phase indicating that the training trials improved thigmotaxis behavior of mice (Fig. 2d). Finally, APP$^{NL-G-F/NL-G-F}$ mice exposed to cognitive training in APP-ST and APP-MT groups showed significant ($P < 0.01$) reduction in thigmotaxia as compared to APP-NT group (Fig. 2e). This last result suggests that both single and multi-domain training decreased thigmotaxia in the APP mice.

The probe trial, however, was a different story. Significant main effects of the quadrant ($F_{(3, 31)} = 15.28$, $P < 0.001$), group ($F_{(3, 31)} = 8.982$, $P = 0.002$), and interaction ($F_{(3, 31)} = 9.002$, $P = 0.002$) suggest performance in the probe trial differed amongst the groups. Follow-up with Sidak's multiple comparisons indicated that only control ($P < 0.001$) and APP-MT ($P < 0.05$) groups spent significantly more time in the target quadrant compared to the average of the remaining three quadrants (Fig. 2f). A proximity measure taken during the probe trial was also analyzed and indicated a main effect of group ($F_{(3, 31)} = 16.990$, $P < 0.001$), and follow up comparisons confirmed that the APP-MT animals had closer proximity to the platform during the probe trial than the APP-NT animals ($P < 0.001$). The APP-MT group did not differ from controls or the APP-ST (Fig. 2g). This effect of cognitive training on spatial specificity suggests that there were some subtle effects of single-domain training, but they were not as strong as multi-domain training.

Overall, these results suggest that multi-domain cognitive training reversed the functional impairments in the APP$^{NL-G-F/NL-G-F}$ mice on the MWM, a task consistently shown to be sensitive to HPC dysfunction. One caveat associated with this conclusion is that although it appeared that multi-domain cognitive training improved performance during the acquisition of the water task, the statistics did not support this observation. One possibility is that because four separate groups were included in this experiment, it is more difficult to pull out an effect due to the inherent increase in variability within

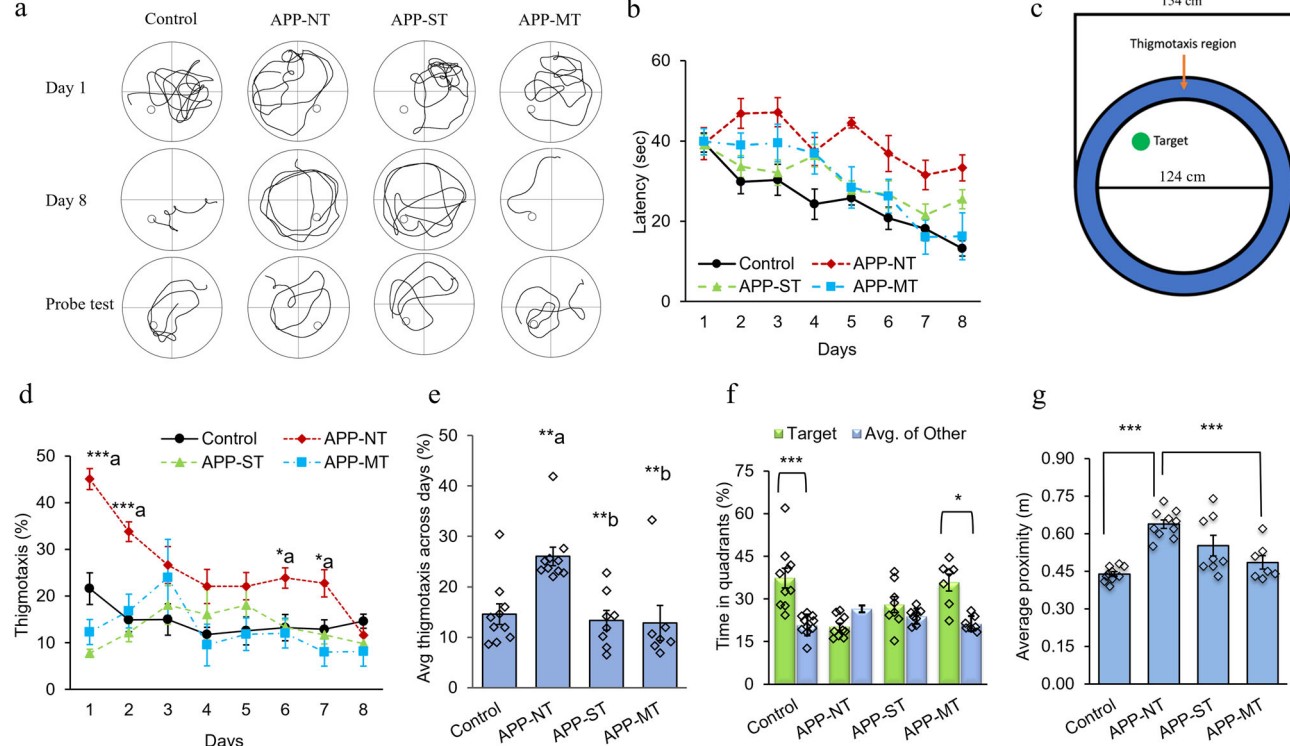

**Fig. 2 Effects of cognitive training on spatial learning and memory functions of 12 months old control and APP$^{NL-G-F/NL-G-F}$ mice in the MWM.**
**a** Representative swim path on Day 1, Day 8, and probe trial for various experimental groups. Overall comparison of experimental groups considering mean latency to find the hidden escape platform during the acquisition phase. **b** Mean latency to find the hidden escape platform during the acquisition phase for various groups. **c** Representative pool schematic showing the size of the outer and inner regions of the pool. Any swimming in the blue region (thigmotaxis area) was counted as time in thigmotaxis. Adopted from the ReadMe in the wtr2100 documentation. **d** Percent thigmotaxis showed by mice on each day during the acquisition trials. **e** Average thigmotaxis (%) across the days shown by mice during the acquisition trials (average of days 1–8 for each group). **f** Percent time spent by mice in the target quadrant and average of other quadrants during the probe trial. **g** Average proximity to the platform during the probe trial. Data is presented as mean ± SEM. Data is presented as mean ± SEM. *$P < 0.05$, **$P < 0.01$, ***$P < 0.001$ are considered as statistically significant. **a**—as compared to the control group; **b**—as compared to the APP-NT group. Control group ($n = 10$)—C57BL/6. APP-NT group ($n = 10$)—APP$^{NL-G-F/NL-G-F}$ mice with no training. APP-ST group ($n = 8$)—APP$^{NL-G-F/NL-G-F}$ mice exposed to single-domain cognitive training (ST). APP-MT group ($n = 7$)—APP$^{NL-G-F/NL-G-F}$ mice exposed to multi-domain cognitive training (MT).

and across groups over the 8 training days. This would not have been an issue with the measures of spatial memory (probe) and specificity (target proximity) as they are single data points collected from each subject on a single probe trial. To assess this idea, we did a different analysis of the acquisition data. We compared the acquisition performance of the different groups on day 1 compared with day 8. Control mice showed a statistically significant ($P < 0.01$) lower escape latency on day 8 ($13.23 \pm 1.91$ s) as compared with day 1 ($39.60 \pm 2.37$ s) indicating normal learning behavior of these mice. In contrast, the 12 months old naïve APP-NT group showed impairments in learning the spatial task as evidenced by a non-significant difference in the escape latency on day 8 ($33.31 \pm 3.25$ s) in comparison to day 1 ($39.35 \pm 3.94$ s). The APP-ST group showed a similar impairment in spatial learning compared to the APP-NT group. That is, no significant difference was observed in escape latency for the APP-ST group between day 1 ($39.05 \pm 3.25$ s) and 8 ($25.52 \pm 5.84$ s) showing no improvement in the learning ability of these mice given this non-pharmacological treatment condition. Interestingly, the APP-MT group showed spatial learning abilities like the control group. Consistent with this claim, the MT prevented the development of learning impairments in APP$^{\text{NL-G-F/NL-G-F}}$ mice as indicated by a significant decrease in the escape latency on day 8 ($16.29 \pm 2.02$ s) when compared with day 1 ($39.88 \pm 3.46$ s; $P < 0.01$). Importantly, we did not find any significant difference in the swim speed of experimental mice (Supplementary Fig. 1).

Taken together, the pattern of effects and statistical analysis show that multi-domain but not single-domain cognitive training improved spatial learning and memory functions rendered dysfunctional in APP$^{\text{NL-G-F/NL-G-F}}$ mice. These effects were particularly pronounced on the spatial probe trial and on spatial specificity measures. Furthermore, either single- or multi-domain cognitive training ameliorated anxiety-related behavior in AD mice as assessed by calculating thigmataxis during the acquisition phase of MWM although multi-domain training was more impactful. These results suggest that multi-domain cognitive training protects a learning and memory network centered on the HPC from AD-related brain changes which normally lead to impairments in the spatial version of the MWM[26,33,34].

*Novel object recognition test.* Pictogram for the object recognition test is shown in Fig. 3a. During the training session, no significant difference was found in exploration time for identical objects 1 and 2 in any experimental groups (Supplementary Fig. 2a). As can be seen in Fig. 3b, during the test session control mice (12 months old) showed normal novel object recognition memory whereas APP-NT mice (12 months old) did not. Specifically, the control mice explored the novel object for significantly longer time when compared to the familiar object (Supplementary Fig. 2b). Furthermore, control mice showed a significantly ($t_{(9)} = -4.282$; $P < 0.01$) higher investigation ratio for the novel object in comparison to the familiar object indicating these mice are spending more time exploring the novel object while the APP-NT group did not ($t_{(9)} = 0.775$; $P > 0.05$), suggesting that the latter group had an object memory impairment (Fig. 3b). APP-ST mice (12 months old) did not show an improvement in memory function evidenced by no significant ($t_{(7)} = -1.070$; $P > 0.05$) difference in the investigation ratio for novel and familiar object (Fig. 3b). In contrast, APP-MT mice (12 months) showed significantly ($t_{(6)} = -4.008$; $P < 0.001$) higher investigation ratio for novel object when compared with familiar object which is also evidenced by significantly more exploration time for the novel object than familiar (Supplementary Fig. 2b). Since this task has been previously shown to be dependent on the perirhinal cortex[35,36] it appears that MT can reverse dysfunction of this medial temporal region following repeated multi-domain training episodes while AD pathology unfolds.

*Fear conditioning test.* The pictogram for the object recognition test is shown in Fig. 3c. Figure 3d shows the results of the fear conditioning to a discrete auditory cue. Control mice (12 months old) showed conditioned fear of an auditory cue and the context associated with a fearful stimulus (foot-shock) while the APP-NT mice (12 months old) did not. One-way ANOVA showed statistically significant differences in percent freezing in the tone ($F_{(3,31)} = 29.279$, $P < 0.001$) as well as context ($F_{(3,31)} = 13.389$, $P < 0.001$) tests among the experimental groups. Consistent with these observations, the least significant difference (LSD) post-hoc analysis indicated that APP-NT mice showed significantly ($P < 0.001$) less percentage of freezing than the control mice to the conditioned tone ($79.67 \pm 2.83\%$ and $45.33 \pm 2.95\%$ for control and APP-NT groups, respectively) and context ($51.50 \pm 3.46\%$ and $24.83 \pm 2.39\%$ for control and APP-NT groups, respectively) (Fig. 3d). APP-MT mice (12 months old) showed an improvement in memory function indicated by significant increases in the percentage of freezing to the tone ($P < 0.01$) and context ($P < 0.001$) in comparison with the APP-NT group (Fig. 3d). The APP-ST group also showed an increase in the percentage of freezing compared to APP-NT mice to the conditioned context ($P < 0.001$), however, no improvements in fear conditioning to the tone was found (Fig. 3d).

This pattern of data suggests that the neural network mediating acquisition of fear conditioning, centered on the amygdala, are compromised in the APP$^{\text{NL-G-F/NL-G-F}}$ mouse model of AD and that repeated multi-domain training protects these brain areas from cognitive impairment.

**Effect of cognitive training on amyloid pathology of APP$^{\text{NL-G-F/NL-G-F}}$ mice.** Photomicrographs of amyloid plaques stained with methoxy-XO4 (green) in the brain of experimental mice are shown in Fig. 4a. We assessed amyloid pathology in different brain regions such as medial prefrontal cortex (mPFC), hippocampus (HPC), retrosplenial area (RSA), perirhinal cortex (PRhC) and cortical amygdalar area (CAA). Therefore, we used brain atlas section A1 (bregma $+1.94$ mm) for mPFC and A2 (bregma $-3.08$ mm) for HPC, RSA, PRhC and CAA. In the present study, MT significantly decreased amyloid load in the brain of APP$^{\text{NL-G-F/NL-G-F}}$ mice, however, ST did not (Fig. 4b). Overall, one-way ANOVA did not reveal the statistically significant differences in percent area for amyloid plaques in brain atlas section A1 ($F_{(2,7)} = 3.071$, $P = 0.110$) and A2 ($F_{(2,7)} = 4.112$, $P = 0.066$) among the experimental groups. However, LSD post-hoc analysis indicated that APP-MT mice showed significantly ($P < 0.05$) less Aβ plaque burden in both A1 and A2 brain sections when compared to the APP-NT group (Fig. 4b). However, the APP-ST group also showed a decrease in Aβ plaques deposition in both brain sections, the differences were not statistically significant when compared to APP-NT mice (Fig. 4b).

We also investigated the deposition of Aβ plaques in various brain regions such as mPFC, HPC, RSA, PRhC, and CAA as these brain regions are involved in various cognitive functions including learning and memory (Fig. 4ci–v). Overall, one-way ANOVA revealed statistically significant differences in percent area for amyloid plaques in HPC ($F_{(2,7)} = 8.599$, $P = 0.013$), RSA ($F_{(2,7)} = 5.432$, $P = 0.038$) and PRhC ($F_{(2,7)} = 9.465$, $P = 0.010$) among the experimental groups but not in mPFC ($F_{(2,7)} = 0.475$, $P = 0.641$) and CAA ($F_{(2,7)} = 3.228$, $P = 0.102$). LSD post-hoc comparison showed that APP-MT mice showed a significant decrease in Aβ plaques burden in HPC ($P < 0.01$, Fig. 4cii), RSA ($P < 0.05$, Fig. 4ciii), PRhC ($P < 0.01$, Fig. 4civ) and CAA ($P < 0.05$, Fig. 4cv) when compared to APP-NT mice. However, ST did not cause a significant change in Aβ deposition in these brain regions of APP$^{\text{NL-G-F/NL-G-F}}$ mice when compared to

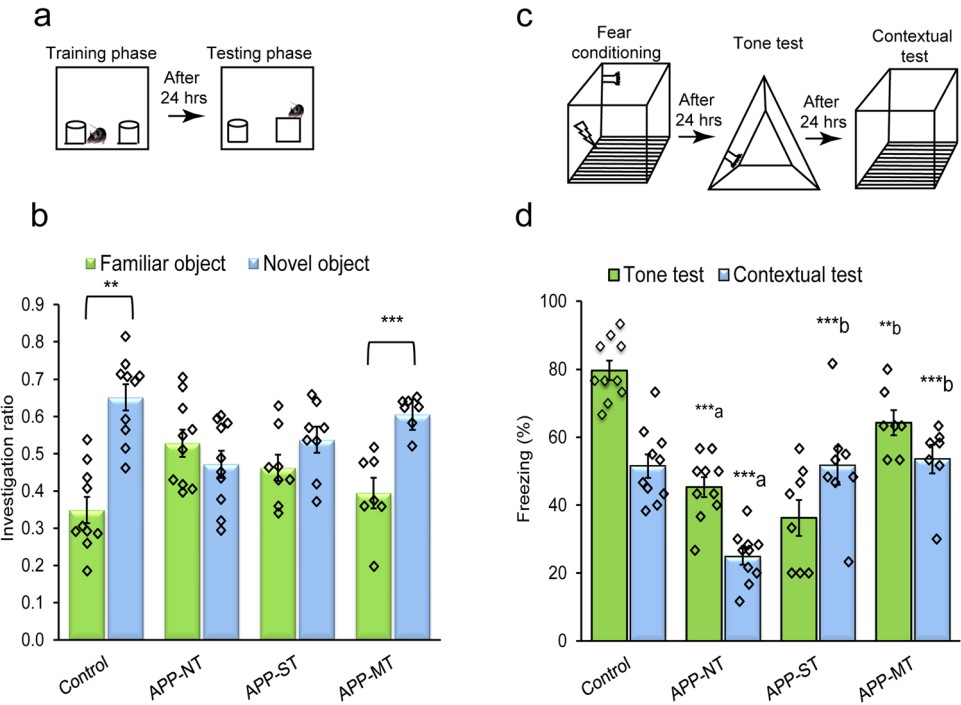

**Fig. 3 Effect of cognitive training on learning and memory functions of 12-month-old APP<sup>NL-G-F/NL-G-F</sup> mice in the novel object recognition (NOR) and fear conditioning tests. a** Schematic representation of the novel object recognition test. **b** Investigation ratio for familiar and novel objects. **c** Schematic representation of the fear conditioning test. **d** Percent freezing for tone and contextual test. Data is presented as mean ± SEM. **$P < 0.01$; ***$P < 0.001$; **a**—as compared with control group. **b**—as compared with the APP-NT group. Control group ($n = 10$)—C57BL/6. APP-NT group ($n = 10$)—APP<sup>NL-G-F/NL-G-F</sup> with no training. APP-ST group ($n = 8$)—APP<sup>NL-G-F/NL-G-F</sup> mice exposed to single-domain cognitive training (ST). APP-MT group ($n = 7$)—APP<sup>NL-G-F/NL-G-F</sup> mice exposed to multi-domain cognitive training (MT).

the APP-NT group (Fig. 4ci–v). Neither of the cognitive training interventions significantly decreased plaque burden in mPFC when compared with the APP-NT group (Fig. 4ci).

Additionally, we assessed amyloid pathology in the medial septum-diagonal band complex (MSDB), a prominent region for cholinergic inputs to various brain regions, especially HPC. Photomicrographs of amyloid plaques stained with methoxy-XO4 (green) in MSDB complex are shown in Fig. 4d. One-way ANOVA indicated statistically significant differences in amyloid plaques number ($F_{(2,7)} = 62.892$, $P < 0.001$) and percent area for amyloid plaques ($F_{(2,7)} = 11.084$, $P < 0.01$) in MSDB among the experimental groups. LSD post-hoc analysis revealed that APP-MT and APP-ST groups showed significant ($P < 0.001$ for MT and $P < 0.01$ for ST) reduction in Aβ plaques number compared to APP-NT group (Fig. 4e). However, the percent Aβ plaques area was only significantly decreased in the APP-MT ($P < 0.01$) group when compared to APP-NT group (Fig. 4f).

The results of repeated cognitive training on amyloid pathology in the APP<sup>NL-G-F/NL-G-F</sup> mice were clear. Amyloid pathology was drastically reduced in brain areas implicated in the learning and memory functions assayed in these same subjects. Namely, amyloid pathology was reduced in the HPC, RSC, PRhC, CAA, and MSDB regions.

**Effect of cognitive training on brain microgliosis of APP<sup>NL-G-F/NL-G-F</sup> mice.** Photomicrographs of microglia stained with IBA-1 (red) in the brains of experimental mice are shown in Fig. 5a. We observed increased microgliosis in the brains of APP-NT mice when compared with the control group. Overall, MT decreased microgliosis in the brains of the APP<sup>NL-G-F/NL-G-F</sup> mice exposed to this treatment in comparison with APP-NT (Fig. 5b). One-way

ANOVA revealed a statistically significant difference in percent immunostained area by IBA-1 in brain atlas sections A1 (+1.94 mm; $F_{(3,11)} = 5.966$; $P = 0.011$) and A2 (−3.08 mm; $F_{(3,11)} = 4.525$; $P = 0.027$) of the experimental groups. With LSD post-hoc analysis, we also found a significant ($P < 0.01$) increase in activated IBA-1 in brain sections A1 and A2 of the APP-NT group as compared to control (Fig. 5b). APP-ST mice had reduced percent microgliosis only in brain section A1 (Fig. 5b). APP-MT mice showed a significant decrease in microgliosis in comparison with mice in the APP-NT group (Fig. 5b).

We also quantified microgliosis in different brain regions which are involved in various cognitive functions. The percent area immunostained by IBA-1 was significantly higher in the APP-NT group in comparison with control mice (Fig. 5ci–v). Overall, one-way ANOVAs revealed statistically significant differences in percent area for activated IBA-1 in mPFC ($F_{(3,11)} = 6.137$, $P = 0.010$), HPC ($F_{(3,11)} = 6.608$, $P = 0.008$), PRhC ($F_{(3,11)} = 5.575$, $P = 0.014$) and CAA ($F_{(3,11)} = 8.746$, $P = 0.003$) among the experimental groups but not in RSA ($F_{(3,11)} = 3.322$, $P = 0.060$). Furthermore, with LSD post-hoc comparison, we also observed significant increases in microgliosis in mPFC ($P < 0.01$, Fig. 5ci), HPC ($P < 0.01$, Fig. 5cii), RSA ($P < 0.05$, Fig. 5ciii), PRhC ($P < 0.01$, Fig. 5civ) and CAA ($P < 0.001$, Fig. 5cv) brain regions of the APP-NT group in comparison with controls. A significant reduction in microgliosis was also observed in mPFC ($P < 0.05$, Fig. 5ci), and CAA ($P < 0.01$, Fig. 5cv) brain regions of the APP-ST group when compared with the APP-NT group. The APP-MT group showed statistically significant reductions of microgliosis in mPFC ($P < 0.05$), HPC ($P < 0.05$), RSA ($P < 0.05$), PRhC ($P < 0.05$), and CAA ($P < 0.01$) brain regions when compared to the APP-NT group (Fig. 5ci–v).

We also assessed microgliosis in the MSDB complex. Photomicrographs of microglia stained with IBA-1 (red) in MSDB are

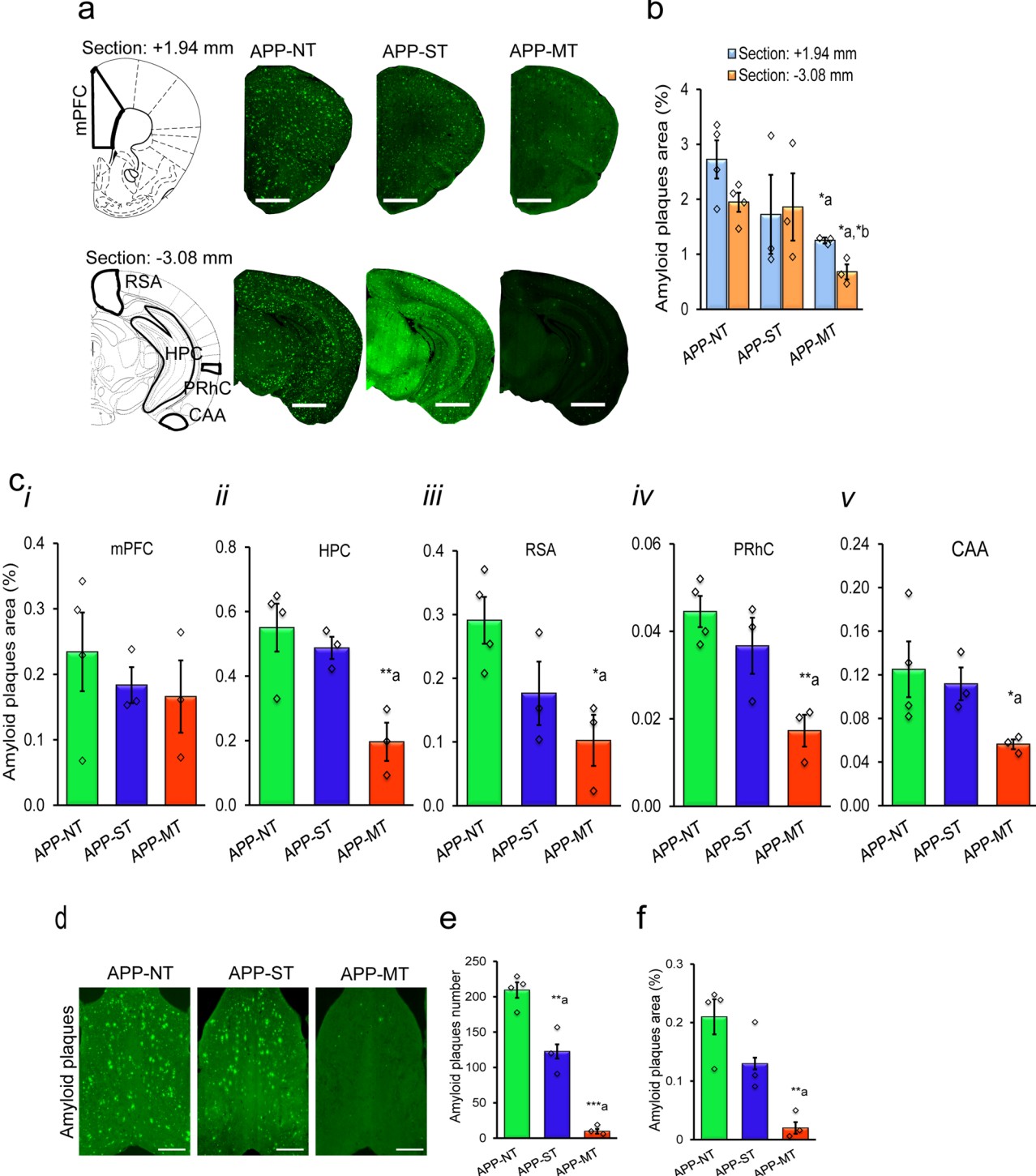

**Fig. 4 Amyloid pathology in brain of 12-month-old APP<sup>NL-G-F/NL-G-F</sup> mice.** a Photomicrographs of amyloid plaques stained with methoxy-XO$_4$ (green) in two brain sections (top: bregma +1.94 mm; bottom: −3.08 mm). **b** Summary results of amyloid plaque area in the brain. **c** Amyloid plaques area in medial prefrontal cortex (mPFC)(i), hippocampus (HPC)(ii) retrospenial area (RSA)(iii), perirhinal cortex (PRhC)(iv), and cortical amygdalar area (CAA)(v). Scale bars represent 1 mm for section +1.94 mm and 2.5 mm for section −3.08 mm. **d** Photomicrographs of amyloid plaques stained with methoxy-XO$_4$ (green) in MSDB complex. **e** Total amyloid plaque number in MSDB complex. **f** Amyloid plaque area in MSDB complex. Scale bar—500 μm for whole sections. Data is presented as mean ± SEM. *$P < 0.05$, **$P < 0.01$, **$P < 0.001$; **a**—as compared to APP-NT group. **b**—as compared to the APP-ST group. APP-NT group ($n = 4$)—APP<sup>NL-G-F/NL-G-F</sup> with no training. APP-ST group ($n = 3$)—APP<sup>NL-G-F/NL-G-F</sup> mice exposed to single-domain cognitive training (ST). APP-MT group ($n = 3$)—APP<sup>NL-G-F/NL-G-F</sup> mice exposed to multi-domain cognitive training (MT).

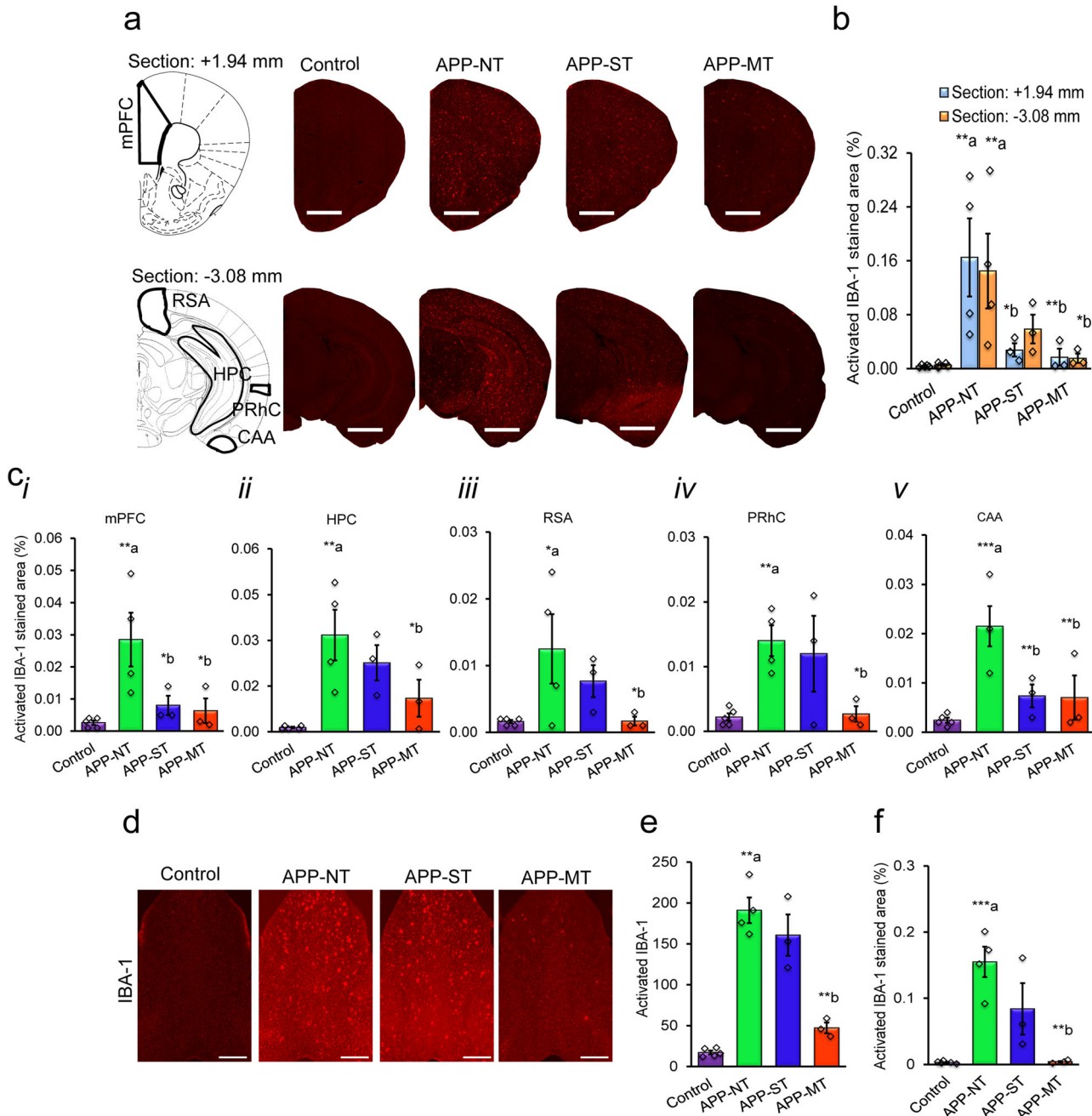

**Fig. 5 Microgliosis in the brain of 12-month-old mice. a** Photomicrographs of activated IBA-1, a marker of microgliosis in the brain. Representative of activated microglia in two brain sections (top: bregma +1.94 mm; bottom: −3.08 mm). Scale bars represent 1 mm for section +1.94 mm and 2.5 mm for section −3.08 mm. **b** Activated IBA-1 immunostained area in brain sections +1.94 mm and section −3.08 mm. **c** Microgliosis in different brain regions of 12 months old mice. Activated IBA-1 immunostained area in the medial prefrontal cortex (mPFC)(i), hippocampus (HPC)(ii), retrospenial area (RSA)(iii), perirhinal cortex (PRhC)(iv), and cortical amygdalar area (CAA)(v). **d** Photomicrographs of immunostaining of activated IBA-1 (red) in MSDB complex. **e** Activated IBA-1 number in MSDB complex. **f** Activated IBA-1 immunostained area in MSDB complex. Scale bar—500 µm for whole sections. Data is presented as mean ± SEM. *$P < 0.05$, **$P < 0.01$, ***$P < 0.001$, **a**—as compared to control group; **b**—as compared to APP-NT group. Control group ($n = 5$)—C57BL/6. APP-NT group ($n = 4$)—APP[NL-G-F/NL-G-F] with no training. APP-ST group ($n = 3$)—APP[NL-G-F/NL-G-F] mice exposed to single-domain cognitive training (ST). APP-MT group ($n = 3$)—APP[NL-G-F/NL-G-F] mice exposed to multi-domain cognitive training (MT).

shown in Fig. 5d. One-way ANOVA indicated the statistically significant differences in activated IBA-1 number ($F_{(3,11)} = 18.146$, $P < 0.001$) and percent area for activated IBA-1 ($F_{(3,11)} = 26.146$, $P < 0.001$) in MSDB among the experimental groups. LSD post-hoc analysis revealed that the APP-NT group showed increases in microgliosis in the MSDB complex evidenced by increased activation of IBA-1 ($P < 0.01$) and percent area stained by IBA-1 ($P < 0.001$) when compared to the control group (Fig. 5e, f). APP-

MT group showed a significant reduction in microgliosis ($P < 0.01$) in the MSDB complex in comparison with the APP-NT group, however, ST intervention did not cause any significant reductions in microgliosis in this brain region (Fig. 5e, f).

In summary, APP[NL-G-F/NL-G-F] mice showed elevated microgliosis in the same brain regions that are functionally compromised in this AD mouse model as revealed via behavioral analysis. Further, the results showed that MT reduces elevated

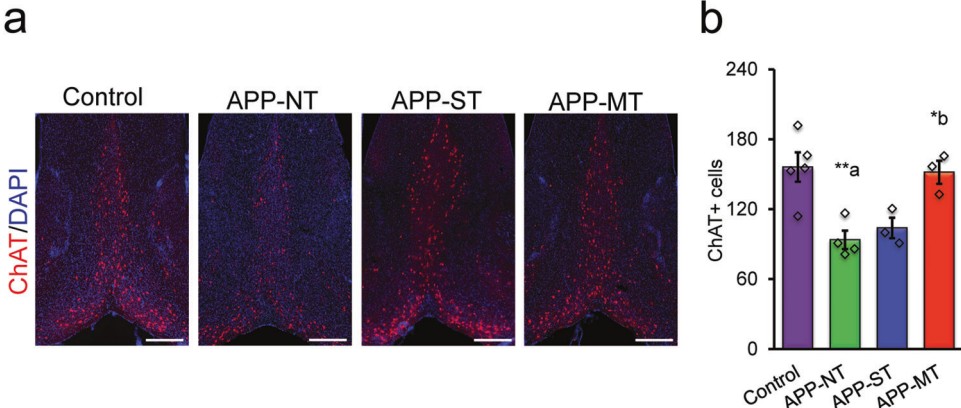

**Fig. 6 Cholinergic function in the medial septum-diagonal band (MSDB) complex of the basal forebrain of 12-month-old mice. a** Photomicrographs of immunohistochemistry staining of choline acetyltransferase (ChAT) in MSDB complex. ChAT—a cholinergic marker stained with monoclonal rabbit anti-ChAT antibody (red), DAPI—stained the nuclei (blue). **b** Quantification of ChAT in MSDB complex. Scale bar—500 μm for whole sections. Data is presented as mean ± SEM. *$P < 0.05$, **$P < 0.01$, **a**—as compared to control group; **b**—as compared to APP-NT group. Control group ($n = 5$)—C57BL/6. APP-NT group ($n = 4$)—APP[NL-G-F/NL-G-F] with no training. APP-ST group ($n = 3$)—APP[NL-G-F/NL-G-F] mice exposed to single-domain cognitive training (ST). APP-MT group ($n = 3$)—APP[NL-G-F/NL-G-F] mice exposed to multi-domain cognitive training (MT).

microgliosis activation levels in these brain regions. However, ST reduces microgliosis only in mPFC and CAA.

**Effect of cognitive training on the cholinergic function of APP[NL-G-F/NL-G-F] mice.** To assess cholinergic function, we performed brain immunostaining for ChAT in the MSDB complex. Photomicrographs of ChAT[+] cells (red) counterstained with DAPI (blue) in the brain of experimental mice are shown in Fig. 6a. Mice in the APP-NT group showed a significant decrease in ChAT positive cells in MSDB complex when compared with control mice (Fig. 6b). Figure 6b shows that the MT intervention prevented the reduction in ChAT positive cells in MSDB complex in comparison with APP-NT mice. Consistent with these observations, one-way ANOVA revealed significant differences between the experimental groups ($F_{(3,11)} = 8.929$, $P = 0.003$). LSD post-hoc analysis revealed a significant ($P < 0.01$) decrease in ChAT-positive cells in the MSDB complex of the APP-NT group ($93.73 \pm 7.88$) when compared with the control group ($156.12 \pm 12.60$, Fig. 6b). The APP-MT group showed a significant reduction in the loss of cholinergic cells ($P < 0.05$) in the MSDB complex in comparison to the APP-NT group (Fig. 6b). Conversely, the APP-ST group did not show significant ($P > 0.05$) difference in number of ChAT positive cells when compared with the APP-NT group (Fig. 6b).

Consistent with our previous work[26], APP[NL-G-F/NL-G-F] mouse model of AD shows drastic reductions in cholinergic neurons in the MSDB. Importantly, the effects of cognitive training on the ascending cholinergic system that innervates the HPC were clear. MT but not ST prevented cholinergic cell loss in the MSDB complex. The resulting enhanced cholinergic tone in HPC might be responsible for at least some of the functional recovery found following MT in this AD mouse model like improvements in spatial learning and memory functions.

## Discussion

In this series of experiments, we sought to assess the potential of cognitive training as a preventative treatment for cognitive impairments and associated brain pathology in AD. Specifically, we tested the effects of repeated exposure to different types of cognitive training in a new-generation mouse model of AD (APP[NL-G-F/NL-G-F]). Male APP[NL-G-F/NL-G-F] mice at 3 months of age were exposed to single-domain or multi-domain cognitive training. ST consisted of a water-based spatial navigation task at 3, 6, and 9. MT consisted of the same spatial navigation task as

well as novel object recognition and fear conditioning. These different forms of cognitive training were given repeatedly at the age of 3, 6, and 9 months of age. At 12 months old all mice were assessed on cognitive tests (NOR, MWM, and FC), and then sacrificed and their brains assessed. APP[NL-G-F/NL-G-F] mice given MT compared to ST showed dramatic improvements in cognitive functions. MT also reduced brain pathology associated with AD including amyloid load and microgliosis as well as preservation of cholinergic function in the brains of APP[NL-G-F/NL-G-F] mice. Interestingly, both ST and MT reduced thigmotaxia. Taken together, this pattern of results provides important experimental evidence that repeated MT can reverse cognitive deficits and associated brain pathology found in Alzheimer's disease.

**Strengths of the current approach.** The present series of experiments is unique in several ways that may represent an advancement in our understanding of the beneficial effects of cognitive training as a preventative treatment for AD. These novel contributions include (1) the use of a new generation mouse model of AD; (2) a comparison of the impacts of repeated single-versus multi-domain cognitive training; (3) a focus on cognitive training in isolation versus in combination with other strategies (environmental enrichment, exercise, pharmacology, etc); (4) the use of various learning and memory tasks as functional assays for different learning and memory networks; (5) the assessment of multiple pathologies associated with AD in many of these same brain areas.

*New generation knock-in mouse model of AD.* More traditional mouse models of AD have overexpressed APP or APP and presinilin1 (PS1) which led to the accumulation of unusual fragments generated by α-secretase, such as C-terminal fragment-β (CTF-β). CTF-β is more toxic than Aβ and CTF-β does not accumulate in human AD brains. A recent study estimates that most neuropathological features of these earlier mouse models are due to artifacts related to APP overexpression[21] and may explain the lack of translational success of all candidate medications tested in clinical trials. We have been using a second-generation AD model developed at the Riken[21], which has a modified APP gene that has a humanized Aβ sequence with three mutations in APP[NL-G-F/NL-G-F]. This mouse model produces a robust age-related spread of Aβ aggregates and cognitive problems with endogenous levels of APP. Our lab has recently characterized the

APP[NL-G-F/NL-G-F] mouse in several experiments and found that these mice display significant Aβ plaque through regions of the neocortex and hippocampus and display cognitive impairments at 6 months, but not 3 months of age[26]. The 6-month-old APP[NL-G-F/NL-G-F] mice also showed increased astrocytosis in the hippocampus, neocortex, medial septum/diagonal band as well as other brain areas. Other brain changes in APP[NL-G-F/NL-G-F] mice include cholinergic and norepinephrine dysfunction. Although further research is required, the demonstration of successful use of repeated MT in the APP[NL-G-F/NL-G-F] mice in reducing both brain pathology and cognitive impairments associated with AD suggests that research investigating preventative treatments using this mouse model of AD might be more translatable to humans because of its similarity to human AD pathology.

*Effects of different forms of cognitive training on various neural networks implicated in learning and memory functions.* In an earlier report, we showed that the APP[NL-G-F/NL-G-F] mouse model of AD, starting from approximately 6 months of age, has learning and memory impairments and the nature of these cognitive deficits indicates that several key memory networks have been rendered dysfunctional[26]. For example, the APP[NL-G-F/NL-G-F] mice showed severe impairments in the acquisition of the spatial version of the MWM. This spatial navigation task is a sensitive assay of a memory network in which the HPC and retrosplenial cortex play central roles[37]. This AD mouse model also shows impairments on the NOR task, a task shown to be dependent on the perirhinal cortex which is a terminal region of the ventral stream of visual processing[38,39]. As well, a neural network centered on the amygdala implicated in aversive and appetitive classical conditioning processes seems compromised as these AD mice were also impaired in cued and context fear conditioning processes[40–42]. In addition to compelling behavioral evidence that these memory networks are dysfunctional in this mouse model of AD, we also showed in the same subjects that these networks show many hallmark pathologies of human AD including amyloid plaques, microglial activation, and cholinergic dysfunction[26].

In the present report, we used this new foundation of knowledge about the APP[NL-G-F/NL-G-F] mouse model of AD and wanted to assess potential preventative treatment approaches. This study focused on indications from epidemiological and clinical studies that cognitive training is a viable preventative approach for this form of age-related cognitive decline. The results clearly showed that these different memory networks were compromised in the APP[NL-G-F/NL-G-F] mouse and multi-domain cognitive training reversed these impairments.

*APP mice, anxiety, and cognitive training.* It has been reported that the APP[NL-G-F/NL-G-F] mouse exhibits an anxiety phenotype[43,44] although there is no consensus on this issue[45–48]. The results of the present study suggest that this mouse model of AD might exhibit anxiety and there does appear to be an anxiety phenotype using other measures including elevated plus maze, light/dark test, and open field that emerge quite early[46,49]. It is important to account for this phenotype because anxiety could impair performance on tasks like the MWM that is independent of learning and memory function and thus simply reducing anxiety via a treatment like cognitive training could improve performance.

Our measure of anxiety in the present study was thigmotaxia (swimming along the pool wall) and it has been shown that this behavior can impair MWM performance[50,51]. It is possible that the multi-domain cognitive training improved MWM performance simply by reducing anxiety. However, we have argued recently[52] that thigmotaxia can emerge for a variety of reasons including altered plasticity and associated learning and memory impairments, anxiety, executive functions, and gonadal

hormones. It is unclear in studies of this type what is driving the increase in thigmotaxia in the APP mice and how cognitive training reverses these impairments.

We envision at least three potential explanations as to what is driving the impairments on the MWM in the APP mice. First, the AD pathology in the APP mice impairs plasticity producing associated learning and memory impairments in the HPC and related neural circuits that render the subjects less confident in finding the escape platform and so they remain near the pool wall. Second, the AD pathology in the APP mice impairs neural networks important for controlling anxiety that are separate from those important for learning and memory functions centered on the HPC neural network. Finally, the hypothesis that we think is most likely is that a combination of learning and memory dysfunction and increased anxiety is driving the impairments in MWM performance in the APP mice. Viewed through this lens, the improved performance on the MWM in the APP mice following multi-domain cognitive training is via both mechanisms, but how?

One hint comes from an interesting pattern emerging from decades of research on the neural basis of learning and memory and parallel work investigating brain systems implicated in anxiety. This work implicates similar neural systems in both learning and memory functions and in controlling fear responses and general anxiety[53–55]. Key brain regions that have been identified include HPC, amygdala, and various parts of the prefrontal cortex. These regions have been implicated in reducing fear and anxiety via learning and memory functions by constraining fear responses to predictive contexts/cues predictive of aversive events[23]. Thus, it is likely that the impacts of multi-domain cognitive training on anxiety are via reduced pathology and dysfunction of the HPC, amygdala, and prefrontal cortex and associated cognitive functions. This analysis is consistent with the claim that voluntary exercise improved learning and memory functions at least partially via decreased anxiety and reductions in various brain pathologies associated with AD in medial temporal lobe brain regions thought to be central in complex neural networks supporting various forms of memory that control fear and anxiety responses.

Our future work using these mouse models of AD will focus on the potential impacts of AD pathology on dorsal and ventral HPC as the latter has been implicated in anxiety and the former in precise spatial navigation. We are also interested in assessing the impacts on orbital frontal cortex as we have found this region to be specifically involved in generalized anxiety but not the learning and memory processes involved in context fear conditioning[55,56].

*Effects of different forms of cognitive training on brain pathology associated with AD.* Along with the powerful cognitive effects of MT, we found associated decreases in Aβ pathology in various brain regions such as MSDB complex, HPC, RSC, PRhC & CCA of 12-month-old APP[NL-G-F/NL-G-F] mice exposed to MT. These brain regions have been implicated in various cognitive functions including learning and memory[37,38,57–60]. The reduction in the amyloid burden in these areas may be responsible for the improved performance on the various assays of learning and memory function.

Interestingly, MT did not reduce the amyloid burden in the mPFC region as it did in the other regions of interest. One explanation for this effect is that the tasks selected would not activate the mPFC as much as it would the three different learning and memory networks in the medial temporal lobes. A variant of the MWM, a one-trial place learning paradigm, could be employed in future work to engage the mPFC sufficiently to improve AD pathology in this cortical region[61].

Glial cell dysfunction observed in the postmortem human AD brain has been reported in various clinical studies[62,63]. In the

present study, we also found that MT reduced microgliosis in APP[NL-G-F/NL-G-F] mice. Interestingly, ST did reduce microgliosis in the mPFC and CAA. The effects of ST on pathology associated with AD in the present experiments are intriguing because although microglial inflammation was reduced in the mPFC, the amyloid burden was not. This points to the possibility that cognitive training can act on multiple mechanisms and via different mechanisms in different brain regions. For example, in the medial temporal lobe, multi-domain cognitive training might reduce amyloid depositions and inflammation emanating from microglial sources. The mPFC might be more sensitive to both ST and MT training which can have positive effects on microglial inflammation but does not reduce Aβ. Further research is required to understand the effects of different forms of cognitive training on different sources of inflammation and any regional-specific effects. We also need to explore what the functional impacts of reduced inflammation in mPFC following multi-domain cognitive training on tasks that tap into the prefrontal system.

The MSDB complex, a part of the basal forebrain, is mainly responsible for cholinergic inputs to HPC and progressive deterioration and/or dysfunction of cholinergic cells in the basal forebrain in aging and neurodegenerative diseases including AD has been reported[64–68]. In a previous study, we also reported the loss of cholinergic neurons in the MSDB complex of APP[NL-G-F/NL-G-F] mice[26]. In the present study, we found that MT prevented the loss of ChAT-positive cells in the MSDB complex in 12-month-old APP[NL-G-F/NL-G-F] mice. However, ST did not rescue the loss of cholinergic neurons. These findings indicate that repeated and varied cognitive training can preserve cholinergic neurons required for various cognitive functions.

One interesting direction for future research on the effects of cognitive training on cholinergic function in the MSDB complex would be to assess HPC acetylcholine content with acetylcholinesterase staining and/or assess the integrity of cholinergic projections. Additionally, previous work has shown that NGF-p75NTR signaling has a critical role in maintaining cholinergic neuron survival so this will be an interesting mechanism to investigate in the future. This evidence would provide additional support for the present findings showing that multi-domain cognitive stimulation enhanced forebrain cholinergic function and a potential brain mechanism for the positive impacts of cognitive training reported in the present study.

Based on the pattern of results reported in this series of experiments, it is highly likely that repeated MT ameliorates cognitive deficits in APP[NL-G-F/NL-G-F] mice via reducing brain amyloid pathology, microgliosis, and preserving cholinergic function.

**Relationship of current findings to the existing literature**. The present report is supported by epidemiological studies in which the importance of life-long learning, as well as mental and physical exercise in aging, has been documented[69]. In addition, correlational human studies indicate that extensive mental exercise seems to provide protection against AD[70,71]. Here, we can also say that MT showed superiority over ST in ameliorating cognitive deficits and pathology in AD. This conclusion is also supported by previous clinical studies showing that an MT program could produce a broader effect on the improvement of overall cognitive functions compared with ST possibly due to cognitive cooperation across different brain processes[72–75]. In line with the present findings, the beneficial effect of MT on neuropsychological outcomes such as delayed memory, naming, visuospatial ability, executive functions, and attention of patients with early-stage AD has been reported recently in a clinical study[76] which supports the use of various cognitive tasks in a repeated manner in the present study. However,

repeated ST did not prevent the development of learning and memory deficits in 12-month-old APP[NL-G-F/NL-G-F] mice on the same task. In contrast to these results, previous studies reported that repeated training in the MWM ameliorates memory deficits in 3xTg-AD mice[14,19]. The discrepancies in the results may be due to the use of the APP-KI mouse (APP[NL-G-F/NL-G-F]) model of AD which is different from the 3xTg-AD mouse model and experimental design.

Several non-pharmacological strategies for reducing AD pathology and associated cognitive impairments have been evaluated including cognitive training, cognitive stimulation and cognitive rehabilitation/enrichment, and enhancing neuroplasticity[16,77–79]. In an interesting systematic review, it has been suggested that cognitive training is most effective when compared to other non-pharmacological approaches[80]. This view is supported by the present results in which we show clear causal evidence that certain forms of cognitive training, in isolation, eliminate cognitive impairments and brain pathology associated with AD. Specifically, we showed that repeated MT but not ST produced these dramatic effects using the APP[NL-G-F/NL-G-F] model of AD.

A big question surrounding this kind of treatment approach for AD is what is the mechanism? Previous research in aging humans suggests that improved learning and memory functions following multi-domain training is complex and probably targets multiple mechanisms including improving synaptic plasticity, increasing blood flow in the brain, improving Aβ clearance, and activating more of the brain[81–83]. Some recent human AD studies show that cognitive training can improve cognition and attentional processes and these changes correlate with changes in the ratios of APP isoforms[84]. Furthermore, one potential mechanism for the region-specific reduction of Aβ plaques could be via a reduction of the number of small-sized Aβ plaques in some regions which raises the possibility that multi-domain training suppresses Aβ production or accelerates its clearance. Future work is required to assess this intriguing possibility.

*Translational potential of the current work*. The issue of the translational potential of biomedical animal work to humans is a controversial one. For example, we think that most would agree that the effort directed at trying to slow or treat AD in humans has not been successful. In our view, part of this failure has occurred because both human and animal researchers refused to embrace the complexity and multifactorial nature of the etiology of AD[85,86].

Moving forward, the ability to significantly ameliorate AD pathology and symptoms will require improvements at all levels (health care funding, research funding, changes in how funding decisions are made, etc) but for human and pre-clinical animal research we feel that several changes will have to be made. First, improvements in mouse models of AD that accurately recapitulate the pathology found in humans. Second, behavioral analysis in many pre-clinical studies varies in quality and sophistication in this area of biomedical research. Third, the strengths and weaknesses of human and animal work need to be emphasized and the need for these different camps to work together meaningfully is fundamental. This kind of approach will improve the translational relevance of the mouse models of AD to impact humans suffering from this debilitating disease.

The design of the present experiments has many of these components including an improved, new-generation genetic mouse model of AD, sophisticated behavioral analysis, associated analysis of brain pathology, and the use of a lifestyle manipulation, and cognitive training that could prove to be effective in human populations. If nothing else, the results suggest that the concept of engaging aged humans in multiple tasks that employ larger swaths of the brain when they are repeatedly exposed to them might be effective in staving off the decline into dementia.

Examples of ways that the treatment approach used in the present experiment could be used in humans are conceivable. AD patients could be repeatedly exposed to three different types of experiences. First, navigate outside to find goals (Sea Hero Quest) that are fun and engaging to activate the HPC network like the MWM in the current experiment. If mobility is an issue virtual reality version of spatial navigation tasks could be used. Second, exposing patients to movies with emotional components (positive and negative) as well as social interaction with family and friends to activate the amygdala like in appetitive and aversive classical conditioning paradigms. Finally, getting the patients to recall objects contained in various homes the individual lived in over the years or exposing them to new objects and families just like the NOR task could be developed.

The results of the present study show that multi-domain cognitive training is a potentially effective non-invasive treatment for human AD patients by reducing brain pathology and improving cognitive function. However, there are a few caveats associated with the experiments reported here. One potential caveat is what the appropriate control for the AD mice used in the present study would be. For these experiments, we used C57BL/6 mice. Although this is the background for the mutant mice it is not a null mutation control. It could be argued that negative littermates of APP$^{NL-G-F/NL-G-F}$ should have been included. We decided to use C57BL/6 as a normal control because when one assesses the research literature using this mouse model of AD many research groups use this approach as well. However, it is important to note that the group that created this mouse model[87] states that APP null mice exhibit the same levels of CTF-β as APP$^{NL-F}$ and APP$^{NL-G-F/NL-G-F}$ mice and that these are the proper negative controls. From our perspective, the use of the APP$^{NL}$ mice as the negative controls might be a mistake as they have one of the mutations associated with AD pathology. Further, these negative mutation controls were not available in our lab at the time. We currently have these mice in the house now and will use them in future experiments to compare with the other groups to see which is the best control.

Another potential caveat is associated with the brain pathology assessments in the present study. We selected 3–5 mice from each group for histopathological assessment which meant that not all the brains from subjects included in the behavioral analysis were assessed. However, there are several reasons to believe that this sampling approach produced an accurate picture of the pathology in the entire groups assessed. First, we have consistently[26,52] found that this number of subjects for pathological assessments is large enough and the patterns of effects have been the same across many studies using this AD mouse model. Regardless, this is important for the reader to know when evaluating the present study.

A final issue is some inconsistencies in the research literature concerning when the behavioral deficits emerge in the APP$^{NL-G-F/NL-G-F}$ mice and what is the nature of these functional changes. For example, we found that the APP$^{NL-G-F/NL-G-F}$ mouse model shows impairments in spatial navigation, fear conditioning, and object recognition at around 6–9 months of age[26] and this is around the time that various AD pathologies significantly increase in various regions of the brain of these mice. However, other work shows that some of these deficits do not appear until later, particularly the MWM deficits. There are several issues concerning these discrepancies. First, the types of tasks used to assess spatial learning and memory are often quite different including Barnes maze, Cross maze, MWM, etc. Second, even though in some cases the same task is used, the methods can be significantly different. For example, we use a modified version of the MWM that uses a distributed trial procedure that gives the subjects fewer trials during each day with longer intertrial

intervals and more training days[26]. This procedure ensures strong quadrant preference, direct swims to the escape platform, and spatial specificity. Further, the size of the pool and cue arrangements in the testing room can potentially impact the complexity of the representation to learn the escape platform position as well as how sensitive the task is to HPC dysfunction. Third, even though the spatial navigation deficit in the MWM seems more difficult to find at 6–9 months of age, other deficits associated with dysfunctional medial temporal lobe structures have been recently reported including other forms of spatial learning and memory and paired associate learning deficits[46,88,89] suggesting that the HPC and related network is compromised at this time point. Fourth, other behavioral changes including cognitive and anxiety deficits have been consistently reported at 6 months suggesting that the AD brain pathology found at this time-point produces functional deficits. Finally, a novel rat APP-KI model has been developed, and those subjects show MWM spatial impairments at 5 months of age[90].

Another potential driver of differences in the onset of functional deficits in this mouse model might be slight differences in the breeding protocols used in different laboratories. What these differences might be is unclear at this time but deserves attention in the future.

Regardless, for the present study, the subjects were tested at 12 months of age or older and many studies find significant anxiety and cognitive functional deficits at this stage of disease progression[46,49,91] including on some of the tasks we use like NOR and FC[92] and spatial deficits are also reported at this age in this mouse model using other tasks like Barnes maze.

Taken together though, it seems prudent to assess functional deficits in this mouse model of AD at 12 months or even later particularly when using the MWM.

*Summary and future directions.* The present experiments provide strong causal evidence that multi-domain cognitive training reverses severe cognitive impairments and associated pathological changes in the brain of a new-generation knock-in mouse model of AD. These effects indicate that repeated cognitive training that engages multiple learning and memory networks will likely be the most effective at reversing neurodegenerative processes. It remains to be seen whether these effects are due in fact to learning different things, or to the influence of general enrichment associated with the different tasks. Exposure to different contexts in the absence of cognitive tasks needs to be compared to multi- versus single-domain cognitive training manipulations to answer this question. Importantly though, these data do demonstrate that the exercise and enrichment garnered from just repeated MWM training is not sufficient to rescue memory (APP-ST). Further, the present results combined with our analysis of the existing research literature suggest that implementing multi-domain cognitive training in combination with other lifestyle factors like voluntary physical exercise[52] or diet could be an effective non-pharmacological approach to prevent or delay the progression of Alzheimer's disease. Future work will be directed at assessing the effects of other lifestyle preventative measures alone or in combination with MT.

## Methods

**Animals and experimental design**. APP knock-in (APP-KI; APP$^{NL-G-F/NL-G-F}$) mice were provided as a gift by the laboratory of Dr. Saido at the RIKEN Center for Brain Science, Japan. APP-KI mice were generated on a C57BL/6 background. A colony of these mice has been maintained at the vivarium at the Canadian Centre for Behavioral Neuroscience, University of Lethbridge. Mice were genotyped and marked using an ear-notching method.

In each cage, four mice were kept in a controlled environment with free access to food and water. All experimental procedures were approved by the institutional animal care committee and performed in accordance with the standards set out by the Canadian Council for Animal Care.

In the present study, we used 35 mice for our experiment. The mice were assigned to four groups including repeated single-domain APP[NL-G-F/NL-G-F] (APP-ST, $n = 8$); repeated multi-domain training APP[NL-G-F/NL-G-F] (APP-MT, $n = 7$); 12 months old no training APP[NL-G-F/NL-G-F](APP-NT, $n = 10$) and 12-month old no training C57BL/6 (control, $n = 10$).

**Single-domain training group**. APP[NL-G-F/NL-G-F] mice ($n = 8$) were given repeated single-domain cognitive training (APP-ST) using the MWM (Fig. 1). For repeated ST, the same mice were trained on the MWM at the ages of 3, 6, 9, and 12 months. The platform position was changed at each time point to rule out the effect of previous training on the improvement in performance with repeated training every three months. At 12 months, mice also underwent object recognition and fear conditioning tests to investigate the effect of repeated training in MWM on the performance of mice in other behavioral tasks. After the completion of behavioral testing, we injected methoxy-XO4 (10 mg/kg) intraperitoneally (i.p.) into mice to stain amyloid plaques in their brain[93]. Then, 24 h after the injection, mice were perfused after a dose of pentobarbital, and their brains were extracted for histology. We also performed immunostaining for microglial and cholinergic function.

**Multi-domain training group**. APP[NL-G-F/NL-G-F] mice ($n = 7$) were exposed to repeated multi-domain cognitive training (APP-MT) (Fig. 1). For the APP-MT group, the same mice were trained on various behavioral paradigms such as the novel object recognition (NOR), standard spatial version of the MWM, and fear conditioning (FC) tests at 3, 6, 9, & 12 months of age. We used the following order for behavioral tests: NOR, MWM, and FC to expose the mice to less to more stress in these neurobehavioral tests. We hypothesized that MT used in this experiment would be more effective than ST used in a single-domain training experiment to ameliorate cognitive dysfunction and AD pathology in the APP-KI mouse model. To rule out the effect of previous training on improvement in performance with repeated training every three months, for NOR, different objects with different geometry, texture, and cleaning solutions to clean object and apparatus were used for each time point; in the MWM test, the platform location was changed at each time point; and for fear conditioning test, different tone, experimental room and cleaning solution to clean apparatus were used for each time point. After the completion of behavioral training, methoxy-XO4 (10 mg/kg, i.p.) was injected in mice to stain amyloid plaques in their brain[93]. Then, 24 h after the injection, the mice were perfused, and their brains were extracted for histology. We also performed immunostaining for microglial and cholinergic function.

To control for the effects of time elapse, that would occur following repeated MT and ST every three months, which might contribute to changes in cognition and AD-related pathology, naïve groups of 12-month-old APP[NL-G-F/NL-G-F](APP-NT) and C57BL/6 were also included in this study. The controls used in this study were from a larger scale study assessing the impacts of cognitive training (the present experiments) or exercise[52] in the APP[NL-G-F/NL-G-F] model. Importantly, all these subjects were run together but it was decided that the study was too large to explore in a single manuscript.

**Behavioral tasks**. To assess the cognitive function of the experimental mice, we performed the Morris water task, novel object recognition, and fear conditioning tests. The behavioral analysis was performed by an experienced researcher blinded to the experimental groups or genotypes of mice.

**Morris water task (MWM)**. MWM experiments were conducted as reported previously[26,94]. In brief, the pool was filled with water, made opaque by adding non-toxic white paint, and the water temperature was maintained at $22 \pm 1\,°C$. The water tank was divided into four quadrants. A circular platform was kept 0.5–1 cm below the water surface in any one quadrant. Three distinct cues of different geometry were placed around the tank. On each of the eight acquisition days, mice received four training trials from each quadrant in a distributed manner. The trial was completed successfully once the mouse found the platform or 60 s had elapsed. If the mouse failed to find the platform on a given trial, the mouse was guided to find the platform. After finding the platform or aided placement, the mouse remained on the platform for 10 s. Mice received full training again at each time-point. Data were recorded using an automated tracking system (HVS Image Hampton, U.K.) and analyzed by wtr2100 Software. The latency to find the platform was used as an indicator of the spatial learning ability of the mice. The swim speed was analyzed to rule out the involvement of motor function as a confounding factor. The Thigmotaxis behavior of mice during the acquisition phase was also analyzed as described previously (Fig. 2c)[52]. A single probe trial was conducted on the ninth day to assess spatial memory performance. The data collected during the probe trial were analyzed measuring the time spent by mice in the target quadrant.

**Novel object recognition test (NOR)**. The advantage of using this test is that it does not require external motivation, reward, or punishment. It is based on the fact that mice will explore novel items. When mice are made familiar with two similar objects during the training day, they will spend more time exploring a novel object on a subsequent test day when in a familiar environment. This pattern of behavior clearly indicates that the mice formed memories of the objects during training and noticed the presence of a novel object during testing[35]. The test was performed as described previously[26,95]. A white, plastic, square box was used for the object recognition test. Briefly, mice were brought from their home cage to the experimental room and familiarized to the testing box for 5 min daily for 2–3 days. Training was conducted 24 h after the last acclimatization day. Two familiar objects were cleaned with 70% isopropyl alcohol to mask any previous odor cues and allowed to dry completely. A video camera was used to record the mice's behavior for further analysis. The mice were put in the square box and allowed to explore the familiar objects for 10 min. After 24 h of training, a test session was conducted. In this session, one of the familiar objects was replaced with a novel object with different geometry and texture. These objects were cleaned with 70% isopropyl alcohol and allowed to dry completely. Mice were individually placed in the testing box for 5 min to explore the objects and a recording of their behavior was made for each mouse. After each mouse completed the test, feces were removed from the testing box. The objects were also wiped with 70% isopropyl alcohol to mask the odor cues after each mouse. The data was analyzed by measuring the exploration time for familiar and novel objects during training and testing days. The investigation ratio was calculated for each group.

**Fear conditioning test (FC)**. This test is conducted to assess the amygdala and hippocampus-associated memory in rodents. Fear conditioning was conducted as described in previous studies[26,95]. FC was conducted in an acrylic square box. A video camera was

used to record the mice's behavior for further analysis. The floor of the chamber consisted of stainless-steel rods. These rods were connected to a shock generator for the delivery of a foot-shock. A speaker was used to deliver a tone stimulus. Prior to conditioning, the chamber was cleaned with a 1% Virkon solution to mask any previous odor cues. On the conditioning day, mice were brought from their home cage into a testing room and allowed to sit undisturbed in their cages for 10 min. Mice were then kept in the conditioning square box and allowed to explore for 2 min before the onset of the tone (20 s, 2000 Hz). In the delay conditioning test, a shock (2 s, 0.5 mA) was given in the last 2 s of tone duration. Mice received five delayed conditioning trials, each separated by a 120-s intertial interval (ITI). The mice were taken from the conditioning chambers 1 min after the last shock and returned to their home cages. Before the tone test, the triangular chamber was cleaned with a 70% isopropyl solution. After 24 h, the tone test was conducted in a triangular chamber that was geometrically different than the conditioning chamber to assess conditioning to the tone in the absence of the training context. For the tone test, three 20-s tones were given after a 2-min baseline period. Each tone presentation was separated by a 120-sec ITI. The freezing response was measured using a time sampling procedure in which an observer scored the presence or absence of the freezing response for each mouse at every 2 s interval. Then, for the context test, 24 h after the tone test, mice were placed back in the original conditioning box for 5-min. During this test, freezing was scored at every 5 s interval. Data was transformed into a percent freezing score by dividing the number of freezing observations by the total number of observations and multiplying by 100. In each conditioning and context procedure, the box was cleaned with 1% Virkon; the tone procedure box was cleaned with 70% isopropyl after each mouse trial to mask any odor cues left by the previous subject and discern context.

**Histology.** After completion of behavioral tests, mice were injected with methoxy-XO4 (10 mg/kg, i.p.) as described in a previous study[93]. Twenty-four hours after injection, mice were transcardially perfused with phosphate-buffered saline (PBS) followed by 4% paraformaldehyde (PFA). Brains were extracted, and post-fixed for 24 h in 4% PFA at 4 °C. Brains were then transferred to 30% sucrose after rinsed with PBS. The histology analysis was performed by an experienced researcher blinded to the experimental groups or genotypes of mice. For histology, 3–5 mice were randomly selected and three sections for each brain region were analyzed, and the data was then averaged. We and others have used these subject numbers and this approach provides sufficient power for statistical analysis[52].

**Quantification of amyloid plaques in mice brain.** We assessed amyloid pathology in different brain regions such as the medial prefrontal cortex (mPFC), hippocampus (HPC), retrosplenial area (RSA), perirhinal cortex (PRhC), and cortical amygdalar area (CAA). Therefore, we used brain atlas section +1.94 mm for mPFC and atlas section −3.08 mm for HPC, RSA, PRhC, and CAA[96]. Fixed brains from mice injected with methoxy-XO4 were frozen and coronally sectioned at 40 μm using a freezing microtome. Brain sections were put on the slide and air dried for 15–20 min. Later, sections on the slides were washed twice with tri-buffered saline (TBS) for 5 min each. Then, brain sections were covered with coverslips with Vectashield H-1000 (Vector Laboratory). Finally, whole slides were imaged using NanoZoomer microscope with 20x objective magnification (NanoZoomer 2.0-RS, HAMAMTSU, JAPAN). The images were analyzed using ImageJ (US National Institutes of Health, Bethesda, Maryland) and Ilastik software[97]. Three sections for each brain region were analyzed and then data were averaged. This software automatically gives the plaque number and size corresponding with each discrete plaque. The number of plaques was quantified according to the plaque size (less than or more than 4 μm). In line with the previous study, 87% of newly generated plaques are small, having a radius of <4 μm[98]. ImageJ (US National Institutes of Health, Bethesda, Maryland) was used to determine the plaque area.

**Immunohistochemistry for microglial and choline acetyl-transferase.** Brains were serially sectioned coronally at 40 μm on a freezing microtome. Immunohistochemical procedures for IBA-1 and ChAT were performed as previously described[21,26,95]. In brief, sections were fixed on positively subbed slides. Brain sections on the slides were washed in TBS, and then, blocked for 2 h in TBS containing 0.3% Triton-X and 3% goat serum. The sections were incubated for 24 h in primary antibody (prepared in TBS with 0.3% Triton-X) at room temperature in a dark humid chamber on the shaker. The following primary antibodies were used: rabbit anti-ChAT (monoclonal, Abcam, ab178850, 1:5000); Anti-Iba1 (Rabbit, SAF4318, 019-19741, Wako). Following incubation, three 10-min washes were done, and sections were again incubated with secondary antibody [goat anti-rabbit-alexa-594 (IgG (H+L), A11037, Invitrogen, 1:1000 for ChAT & IBA-1] for 24 h. Following incubation with a secondary antibody, three 10-min washes were done. Then, only ChAT sections were incubated with DAPI (1:2000 of the 20 μg/ml stock in TBS) for 45–60 min. After incubation, a single 5 min wash was given. Finally, sections on the slides were cover-slipped with Vectashield H-1000 (Vector Laboratory), and the slides were sealed with nail polish. Whole slides were then imaged using NanoZoomer microscope with 20x objective magnification (NanoZoomer 2.0-RS, HAMAMTSU, JAPAN). The images were analyzed using ImageJ and Ilastik software[97]. Three sections for each brain region were analyzed and then data were averaged.

**Statistics and reproducibility.** The statistical analyses were conducted using the SPSS statistical software package, version 22.0. Results are given as mean ± SEM. A mixed model ANOVA with repeated measures along with Sidak's multiple comparisons was used to find statistically significant differences across the days during the acquisition phase of MWM. A paired t-test was used to measure significant differences between the target and the average of other quadrants during the probe trial. Fisher's LSD was used as a post-hoc comparison test to find significant differences between the groups. One-way ANOVA with LSD as post-hoc was used to find significant differences between the experimental groups for fear conditioning and histology. Additionally, a paired t-test was used to find statistically significant differences in the object recognition test. A P-value < 0.05 was considered statistically significant.

**Reporting summary.** Further information on research design is available in the Nature Portfolio Reporting Summary linked to this article.

### Data availability
The source data for all figures and supplementary figures can be found in Supplementary Data 1. The raw data generated and analyzed in this study are available from the corresponding authors upon reasonable request.

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

## Acknowledgements

This work was supported by Natural Sciences and Engineering Research Council of Canada (NSERC) Discovery Grant #40352, #06347, and #03857 to M.H.M., R.J.M., and R.J.S. respectively, Alberta Innovates (M.H.M.), Alberta Alzheimer Research Program (M.H.M.), Alzheimer Society of Canada (M.H.M., R.J.M.), Alberta Prion Research Institute (M.H.M., R.J.S.), and Canadian Institute for Health Research (M.H.M., R.J.M.). We thank Dr. Takashi Saito and Prof. Takaomi C Saido from "The Laboratory for Proteolytic Neuroscience RIKEN Center for Brain Science, Wako-shi, Saitama, Japan" for providing the APPNL-G-F/NL-G-F mice as a gift. We also thank Di Shao for animal breeding.

## Author contributions

J.M., M.H.M., and R.J.M. designed and conceptualized the experiments. J.M., S.H., S.G.L., and S.H.D. performed the behavioral experiments. J.M. analyzed the behavioral data. J.M. performed the immunohistochemistry. J.M. & H.K. analyzed the immunohistochemistry data. J.M., N.S.H., R.J.M., and M.H.M. wrote the manuscript, which all authors commented on and edited. M.H.M., R.J.M., and R.J.S. provided the resources. M.H.M. and R.J.M. supervised the study. All authors approved the final version of the manuscript.

## Competing interests

The authors declare no competing interests.
