## [Peer Review File · Communications Biology]

Reviewers' comments:

Reviewer #1 (Remarks to the Author):

This is a carefully executed study, nicely illustrated, that demonstrates more beneficial effects of multidomain versus single-domain training in APP NL-G-F knock-in mice. Experienced research group. It all seems very reliable, and data are clearly presented.

Criticisms:

(1) I have a major conceptual problem with this study because it refers to single/multiple domains of behavioral performance without clearly explaining what this means. Some authors have done PCA or factor analysis on the variables in a behavioral battery. Also, the groups of Lipp and Wolfer performed factor analysis on MWM data and identified different factors there. Does multidomain mean comprising different PCA factors for instance, or does it just mean different tests? It feels like a fancy way to indicate that one group just did a single test, whereas the other group did a couple of behavioral tests. The conceptual basis of calling it multidomain is unclear. In fact, their beneficial effect of so-called multidomain training might boil down to doing more, moving more, being handled more, more complex environmental stimulation, etc.

(2) The introduction talks a lot about Alzheimer patients and how they benefit from remaining active. One-to-one translation of mouse work to patients is exceedingly difficult. A lot of the clinical information is only superficially relevant for these mouse studies. These paragraphs should be shortened and made more relevant to the present study. The third paragraph of the introduction compares extensively between findings in AD patients and AD mouse models, but without proper referencing.

(3) The behavioral profile of the APP NL-G-F knock-in model is only briefly introduced in §4, Introduction. Other authors reported only subtle defects in these mice – some have found none at all. The authors should relate their behavioral observations to those of others in this particular model. In general, there should be more background about the model in the introduction.

(4) It is not indicated whether experimenters were blinded to the genetic status of the mice.

(5) Very little rationale with regard to the choice of the 3 behavioral tests. MWM is notoriously robust within as well as between labs, NOR much less so.

(6) There are some idiosyncrasies in their MWM testing that should be explained further. Water temp. of 22°C is very cold, 60s cut-off is rather short. Swim speed as a motor function parameter is outdated. Swim speed also has cognitive elements, mice swim faster as they start to master the task. To isolate the cognitive aspects of the task, it is preferable to determine search strategies or measure more sophisticated variables. Use the more common acronym MWM instead of MWT, MWM test, MWM task (also, conflicting use of terms in fig. 1).

Reviewer #2 (Remarks to the Author):

In this article, the authors investigated the effects of cognitive training for memory impairments and brain pathology in Alzheimer's disease (AD) using a knock-in mouse model. They found that multi-domain cognitive training prevented more effectively cognitive deficits in AppNL-G-F mice in several behavioral tests. In addition, A β deposition and microgliosis were ameliorated in memory-related brain regions, and cholinergic neuron positive for ChAT were more preserved in AppNL-G-F mice with multi-domain cognitive training. Their results may provide evidence that cognitive training or cognitive enhancement have beneficial effects to prevent cognitive decline in AD patients. The study is well designed and conducted, though the reviewer raises several suggestions below to revise the manuscript for publication.

1. In the Morris water task, AppNL-G-F mice with single- or multi-domain cognitive training exhibit less thigmotaxic behavior compared with the mice with no training. As the authors described, thigmotaxic behavior is shown to reflect animal's emotional state, whereas it may be associated with alterations of search strategy to reach the platform. Can the authors discuss these points further? Since some reports says that AppNL-G-F mice exhibit more anxiolytic behaviors in other tests (e.g. open field, elevated plus maze), for the author's conclusion, this reviewer thinks that it is better to evaluate anxiety-like behavior in AppNL-G-F mice with cognitive training using other tests.

2. As the authors mentioned, multi-domain cognitive training also improved the performance of 12-month-old AppNL-G-F mice in the novel object recognition test. The mice exposed to single-domain training showed low levels of exploring objects compared to other 3 groups. This reviewer thinks that low exploratory activity may result in lower discrimination index. Did these mice show similar levels of exploration during the habituation?

3. Histological analyses revealed that multi-domain cognitive training ameliorate A β deposition in AppNL-G-F mice, however, there were some regional differences for the phenotype. As the data presented, multi-domain cognitive training can ameliorate A β deposition in some regions such as the hippocampus and perirhinal cortex, but not in the medial prefrontal cortex. Can the authors discuss for the regional differences that multi-domain cognitive training affect the brain A β pathology? Indeed, decreased number of small-sized A β plaques in some regions can raise the possibility that multi-domain training suppress A β production or accelerate its clearance. Can the authors discuss how cognitive training affect A β metabolism? As a minor point related with the data, this reviewer recommends replacement of the image showing A β deposition at section (-3.08mm) in AppNL-G-F mice with single-domain cognitive training.

4. AppNL-G-F mice with single- and multi-domain training exhibit reduced microgliosis in the medial prefrontal cortex as shown by Iba1 staining. It is some curious because these mice have similar levels of A β deposition in this region compared with the mice with no training. Can the authors further discuss these points?

Also, this reviewer wonders if multi-domain cognitive training ameliorates astrocytosis as well. Since astrocyte has known to be involved in various homeostatic functions including neuronal support, the data would be of importance to speculate the underlying mechanisms that cognitive training can ameliorate behavioral deficits or brain pathology.

5. Interestingly, AppNL-G-F mice exposed to multi-domain cognitive training exhibited very few A β plaques in the MSDB complex than other brain regions. And, they also preserved the relative number of ChAT-positive cholinergic neurons. Do the authors have any data to assess brain acetylcholine content or cholinergic projections to support that cognitive stimulation enhance forebrain cholinergic function? And, can the authors discuss underlying mechanisms that cognitive training

preserve cholinergic system? Previous studies show that NGF-p75NTR signaling has a critical role for maintaining cholinergic neuron survival.

6. As the authors described, 3 behavioral tests (Morris water task, novel object recognition test, and fear conditioning test) were performed for cognitive training and tests. Some behavioral tests are established for assessing cognitive function and memory in mice, so are there any reasons why the authors included these 3 tests?

7. What order did the authors conduct the behavioral tests in multi-domain cognitive training? Some behavioral tests are stressful to mice. In these 3 tests It may be the order that novel object recognition, Morris water task, and fear conditioning to arrange from less to more stressful in these 3 tests. If possible, the information may be included in Figure 1.

8. It may be more helpful for readers in other fields to include the information for animal numbers used in each behavioral and histological analysis in each Figure legends.

Reviewers' comments

We would like to thank the reviewers for their helpful and fair comments on our manuscript. The reviewer's comments are indicated below in normal black font. Our responses are in purple font and the changes in the manuscript text are indicated in green.

Reviewer #1 (Remarks to the Author)

This is a carefully executed study, nicely illustrated, that demonstrates more beneficial effects of multidomain versus single-domain training in APP NL-G-F knock-in mice. Experienced research group. It all seems very reliable, and data are clearly presented.

Comment 1: I have a major conceptual problem with this study because it refers to single/multiple domains of behavioral performance without clearly explaining what this means. Some authors have done PCA or factor analysis on the variables in a behavioral battery. Also, the groups of Lipp and Wolfer performed factor analysis on MWM data and identified different factors there. Does multidomain mean comprising different PCA factors for instance, or does it just mean different tests? It feels like a fancy way to indicate that one group just did a single test, whereas the other group did a couple of behavioral tests. The conceptual basis of calling it multidomain is unclear. In fact, their beneficial effect of so-called multidomain training might boil down to doing more, moving more, being handled more, more complex environmental stimulation, etc.

Reply: We really appreciate this issue raised by the reviewer. This was a glaring omission on our part, and we apologize. In the present study, we have not done PCA or factor analysis on the variables in a behavioral battery. To clarify, we used Morris water task (MWM) at different time points (3, 6, 9 & 12 months) for single-domain cognitive training (Fig. 1). We used MWM, object recognition and fear conditioning tests at different time points (3, 6, 9 & 12 months) for multi-domain cognitive training (Fig. 1). A new section has been added to the introduction on page 4. Below finds the added paragraph.

“Our goal in designing these experiments was to select a task that would preferentially and repeatedly activate one learning and memory network (single domain training) and compare the effects of this brain activation during disease progression to subjects exposed to multiple tasks that would preferentially and repeatedly activate three different learning and memory networks (multi-domain training). There is a large body of evidence in mice, rats, monkeys, and humans that there are multiple learning and memory networks that are centered on key brain regions. The hippocampus is a central player in a learning and memory network involved in episodic memory. The amygdala is a key player in a network important for emotional learning and memory functions. Finally, the perirhinal cortex is a key component of a learning and memory network important for object memory (for review of this literature see McDonald et al., 2017). The hypothesis is that repeatedly activating multiple networks would be more beneficial than a single network.”

Reference

- McDonald, R.J., Hong, N.S., and Devan, B.D. (2017) Interactions among multiple parallel learning and memory systems in the mammalian brain. Chapter published in Howard Eichenbaum (Ed). Learning and Memory: A Comprehensive Reference (2nd Edition).

Comment 2: The introduction talks a lot about Alzheimer patients and how they benefit from remaining active. One-to-one translation of mouse work to patients is exceedingly difficult. A lot of the clinical information is only superficially relevant for these mouse studies. These paragraphs should be shortened and made more relevant to the present study. The third paragraph of the introduction compares extensively between findings in AD patients and AD mouse models, but without proper referencing.

Reply: We agree with the reviewer that some of the references used in the introduction were not the best for supporting the claims that we were making. In multiple cases (citation 12, 13 and 15) these citations were experiments assessing environmental enrichment in mouse models of AD which were not appropriate in this section. We have dropped these citations and maintained only the ones relevant to the impacts of cognitive training in humans suffering with AD and cognitive training in mouse models of AD. We thank the reviewer for carefully checking this important section of the paper.

We also agree with the reviewer on the issue of translation of animal work to humans. The effort directed at trying to slow or treat AD in humans has been disappointing. In our view, part of this failure has occurred because both human and animal researchers refused to embrace the complexity and multifactorial nature of the etiology of AD (see McDonald et al., 2010; Gidyk et al., 2015).

We have added a new section on page 24 in the discussion entitled “Translational potential of the current work” discussing these issues which are highlighted in green color in revised manuscript:

In our view, the ability to significantly ameliorate AD pathology and symptoms will require improvements at all levels (health care funding, research funding, changes in how funding decisions are made, etc) but for human and pre-clinical animal research we feel that several changes will have to be made. First, improvements in mouse models of AD that accurately recapitulate the pathology found in humans. Second, behavioural analysis in many pre-clinical studies varies in quality and sophistication in this area of biomedical research. Third, the strengths and weaknesses of the human and animal work need to be emphasized and the need for these different camps to work together meaningfully is fundamental.

This kind of approach will improve translational relevance of the mouse models of AD to impact humans suffering from this debilitating disease.

The design of the present experiments has many of these components including an improved, new generation genetic mouse model of AD, sophisticated behavioural analysis, and associated

analysis of brain pathology in parallel, and the use of a lifestyle manipulation, cognitive training, that could prove to be effective in human populations. If nothing else, the results suggest that the concept of engaging aged humans in multiple tasks versus one that engage larger swaths of the brain when they are repeatedly exposed to them will be effective in staving off the decline into dementia.

Examples of ways that the treatment approach used in the present experiment could be used in humans are conceivable. The AD patients could be repeatedly exposed to three different types of experiences. First, navigating outside to find particular goals (Sea Hero Quest) that are fun and engaging to activate the HPC network like the MWM in the current experiment. If mobility is an issue virtual reality version of spatial navigation tasks could be used. Second, exposing patients to shows with emotional components (positive and negative) as well as social interaction with family and friends to activate the amygdala like in appetitive and aversive classical conditioning paradigms used in rodents. Finally, getting the patients to recall objects contained in various homes the individual lived in over the years or exposing them to new objects and familiar just like the Novel Object Recognition task could be developed.

References

- Gidycz, D. C., Deibel, S. H., Hong, N. S. & McDonald, R. J. Barriers to developing a valid rodent model of Alzheimer's disease: from behavioral analysis to etiological mechanisms. *Front. Neurosci.* 9, 245 (2015).
- McDonald, R. J., Craig, L. A. & Hong, N. S. The etiology of age-related dementia is more complicated than we think. *Behav. Brain Res.* 214, 3–11 (2010).

Comment 3: The behavioral profile of the APP NL-G-F knock-in model is only briefly introduced in page 4, Introduction. Other authors reported only subtle defects in these mice – some have found none at all. The authors should relate their behavioral observations to those of others in this particular model. In general, there should be more background about the model in the introduction.

Reply: We thank the reviewer for useful suggestion. Additionally, as per the reviewer's suggestion, we have also incorporated the following lines under the heading "Introduction" on page 4 which are highlighted in green color in revised manuscript:

First generation transgenic mouse models of AD overexpressed APP and APP fragments which may be responsible for artificial phenotypes in mice (Hsiao et al., 1996, Saito et al., 2014). Saito et al. developed a single APP knock-in (APP-KI) mouse model for AD to overcome problems with APP overexpression (Saito et al., 2014). APP^{NL-G-F} mice showed typical A β pathology, neuroinflammation cholinergic dysfunction, and neurobehavioral impairment (Saito et al., 2014; Masuda et al., 2016; Mehla et al., 2019). This mouse model is being used more commonly than other APP-KI line as it develops A β pathology faster (Saito et al., 2014) and can be used to study various downstream mechanisms such as neuroinflammation (Shirotani et al., 2019; Sobue et al., 2021), pericyte signaling (Nortley et al., 2019), oxidative stress (Urano et al., 2020), tau

propagation (Saito et al., 2019), and spatial memory impairment (Masuda et al., 2016; Mehla et al., 2019; Sutoko et al., 2021). This is the mouse model of AD that we have selected for the present study.

References

- Hsiao, K. et al. Correlative memory deficits, Abeta elevation, and amyloid plaques in transgenic mice. *Science* 274, 99–102 (1996).
- Masuda, A. et al. Cognitive deficits in single App knock-in mouse models. *Neurobiol. Learn. Mem.* 135, 73–82 (2016).
- Mehla, J. et al. Age-dependent behavioral and biochemical characterization of single APP knock-in mouse (APP^{NL-G-F/NL-G-F}) model of Alzheimer’s disease. *Neurobiol. Aging* 75, 25–37 (2019).
- Nortley, R. et al. Amyloid β oligomers constrict human capillaries in Alzheimer’s disease via signaling to pericytes. *Science* 365, (2019).
- Saito, T. et al. Humanization of the entire murine Mapt gene provides a murine model of pathological human tau propagation. *J. Biol. Chem.* 294, 12754–12765 (2019).
- Saito, T. et al. Single App knock-in mouse models of Alzheimer’s disease. *Nat. Neurosci.* 17, 661–3 (2014).
- Shirotani, K. et al. Aminophospholipids are signal-transducing TREM2 ligands on apoptotic cells. *Sci. Rep.* 9, 7508 (2019).
- Sobue, A. et al. Microglial gene signature reveals loss of homeostatic microglia associated with neurodegeneration of Alzheimer’s disease. *Acta Neuropathol. Commun.* 9, 1 (2021).
- Sutoko, S. et al. Early identification of Alzheimer’s disease in mouse models: Application of deep neural network algorithm to cognitive behavioral parameters. *iScience* 24, 102198 (2021).
- Uruno, A. et al. Nrf2 Suppresses Oxidative Stress and Inflammation in App Knock-In Alzheimer’s Disease Model Mice. *Mol. Cell. Biol.* 40, (2020).

We agree with the reviewer that one confusing aspect of the functional assessment of this APP-KI mouse model is that there are seemingly conflicting effects of the impacts of the pathology on behaviour. We have added some additional paragraphs on the page 25-26 highlighted in green in the revised manuscript addressing this important issue in the “Caveats” section of the discussion (see below):

“A final issue is some inconsistencies in the research literature concerning when the behavioural deficits emerge in the APP^{NL-G-F} mice and what is the nature of these functional changes. For example, we have found that the APP^{NL-G-F} mouse model shows impairments in spatial navigation, fear conditioning, and object recognition at around 6-9months of age (Mehla et al., 2019) and this is around the time that various AD pathologies significantly increase in various regions of the brain of these mice. However, other work shows that some of these deficits do not appear until later, particularly the MWM deficits. There are several issues concerning these

discrepancies. First, the types of tasks used to assess spatial learning and memory are often quite different including Barnes maze, Cross maze, MWM, etc. Second, even though in some cases the same task is used, the methods can be significantly different. For example, we use a modified version of the MWM that uses a distributed trial procedure that gives the subjects fewer trials during each day with longer intertrial intervals and more training days (Mehla et al., 2019). This procedure ensures strong quadrant preference, direct swims to the escape platform, and spatial specificity. Further, the size of the pool and cue arrangements in the testing room can potentially impact the complexity of the representation to learn the escape platform position as well as how sensitive the task is to HPC dysfunction. Third, even though the spatial navigation deficit in the MWM seems more difficult to find at 6-9 months of age, other deficits associated with dysfunctional medial temporal lobe structures have been recently reported including other forms of spatial learning and memory and paired associate learning deficits [Sakakibara et al. 2018; Saifullah et al., 2020; Whyte et al., 2022] suggesting that the HPC and related network is compromised at this time point. Further, other behavioural changes including cognitive and anxiety deficits have been consistently reported at 6 months suggesting that the AD brain pathology found at this time-point produces functional deficits. Further, a novel rat APP-KI model has been developed, and those subjects show MWM spatial impairments at 5 months of age (Pang et al., 2022).

Another potential driver of differences in onset of functional deficits in this mouse model might be slight differences in the breeding protocols used in different laboratories. What these differences might be is unclear at this time but deserves attention in the future.

Regardless, for the present study the subjects were tested at 12 months of age or older and many studies find significant anxiety and cognitive functional deficits at this stage of disease progression (Auta et al., 2021; Sakakibara et al., 2018; Mizuno et al., 2022) including on some of the tasks we use including NOR and fear conditioning (Wang et al., 2022), and spatial deficits are also reported at this age in this mouse model using other tasks like Barnes maze.

Taken together though, it seems prudent to assess functional deficits in this mouse model of AD at 12 months or even later particularly when using the MWM.

References

- Auta, J., Locci, A., Guidotti, A., Davis, J. M. & Dong, H. Sex-dependent sensitivity to positive allosteric modulation of GABA action in an APP knock-in mouse model of Alzheimer's disease: Potential epigenetic regulation. *Curr. Res. Neurobiol.* 3, 100025 (2022).
- Mehla, J. et al. Age-dependent behavioral and biochemical characterization of single APP knock-in mouse (APP^{NL-G-F/NL-G-F}) model of Alzheimer's disease. *Neurobiol. Aging* 75, 25–37 (2019).
- Mizuno, Y. et al. Deficiency of MTH1 and/or OGG1 increases the accumulation of 8-oxoguanine in the brain of the App^{NL-G-F/NL-G-F} knock-in mouse model of Alzheimer's disease, accompanied by accelerated microgliosis and reduced anxiety-like behavior. *Neurosci. Res.* 177, 118–134 (2022).

- Pang, K. et al. An App knock-in rat model for Alzheimer's disease exhibiting A β and tau pathologies, neuronal death and cognitive impairments. *Cell Res.* 32, 157–175 (2022).
- Saifullah, M. A. Bin et al. Touchscreen-based location discrimination and paired associate learning tasks detect cognitive impairment at an early stage in an App knock-in mouse model of Alzheimer's disease. *Mol. Brain* 13, 147 (2020).
- Sakakibara, Y., Sekiya, M., Saito, T., Saido, T. C. & Iijima, K. M. Cognitive and emotional alterations in App knock-in mouse models of A β amyloidosis. *BMC Neurosci.* 19, 46 (2018).
- Wang, S. et al. Age-Dependent Behavioral and Metabolic Assessment of App NL-G-F/NL-G-F Knock-in (KI) Mice. *Front. Mol. Neurosci.* 15, 909989 (2022).
- Whyte, L. S. et al. Lysosomal gene Hexb displays haploinsufficiency in a knock-in mouse model of Alzheimer's disease. *IBRO Neurosci. reports* 12, 131–141 (2022).

Comment 4: It is not indicated whether experimenters were blinded to the genetic status of the mice.

Reply: The experimenters who performed the behavioural as well as the histological assessment were blind to the genetic status of the mice. We have incorporated the following lines highlighted in green color in revised manuscript:

Under the heading “Behavioral experiments” on page 7

The behavioral analysis was performed by an experienced researcher blinded to the experimental groups or genotype of mice.

Under the heading “Histology” on page 9

The histology analysis was performed by an experienced researcher blinded to the experimental groups or genotype of mice.

Comment 5: Very little rationale with regard to the choice of the 3 behavioral tests. MWM is notoriously robust within as well as between labs, NOR much less so.

Reply: We appreciate the reviewer noticing this omission in the manuscript. This comment is like the one raised in Comment 1 by this same reviewer and as we state in that response these tasks were selected to activate different networks implicated in different forms of learning (McDonald et al., 2017). Specifically, networks centered on the hippocampus (MWM), amygdala (fear conditioning) and perirhinal cortex (object recognition). Interestingly, these regions are also key targets of AD pathology.

We agree with the reviewer that the MWM task is more reliable than the NOR task within and between labs but for our purposes it did not seem to be an issue as we obtained robust NOR scores in our control groups and we have reliably obtained these effects in other studies using these same parameters (Mehla et al., 2018;2019;2022).

References

- McDonald, R.J., Hong, N.S., and Devan, B.D. (2017) Interactions among multiple parallel learning and memory systems in the mammalian brain. Chapter published in Howard Eichenbaum (Ed). Learning and Memory: A Comprehensive Reference (2nd Edition).
- Mehla et al. Dramatic impacts on brain pathology, anxiety, and cognitive function in the knock-in APPNL-G-F mouse model of Alzheimer disease following long-term voluntary exercise. *Alzheimers Res Ther.* 2022 Sep 30;14(1):143.
- Mehla, J. et al. Age-dependent behavioral and biochemical characterization of single APP knock-in mouse (APPNL-G-F/NL-G-F) model of Alzheimer's disease. *Neurobiol. Aging* 75, 25–37 (2019).
- Mehla et al. Gradual Cerebral Hypoperfusion Impairs Fear Conditioning and Object Recognition Learning and Memory in Mice: Potential Roles of Neurodegeneration and Cholinergic Dysfunction. *Alzheimers Dis.* 2018;61(1):283-293.

Comment 6: There are some idiosyncrasies in their MWM testing that should be explained further. Water temp. of 22°C is very cold, 60s cut-off is rather short. Swim speed as a motor function parameter is outdated. Swim speed also has cognitive elements, mice swim faster as they start to master the task. To isolate the cognitive aspects of the task, it is preferable to determine search strategies or measure more sophisticated variables. Use the more common acronym MWM instead of MWT, MWM test, MWM task (also, conflicting use of terms in fig. 1).

Reply: We appreciate the reviewer's comment. It is consistently reported that mice have an increased tendency of floating as compared to rats during training on the MWM test and in pilot studies this is in fact what we find. Therefore, to prevent floating behavior of mice, we keep the water temperature at 22°C to motivate them to find the platform as quickly as possible. To ameliorate effects of the cold temperature on the mice we only gave the subjects 4 trials per day for 8 days, instead of 8 trials per day for 4 days and a long inter-trial interval (20 minutes) was used.

Our paradigm differs in other ways as well and for what we feel are good reasons. Our reading of a significant amount of AD research using mice is that the training parameters normally used for the MWM are not sufficient to obtain clear place learning including spatial specificity of searching in the target quadrant (Mehla et al., 2018).

We were not sure if the reviewer was referring to a 60 second probe trial or the 60 second maximum time for each trial.

Regarding the probe trial, 60 seconds in our view is quite long. In much of our work, we now use 30 seconds, and the data does not differ much (Mehla et al., 2019). Any more than 60 seconds and the subjects can start to give up on the target location and start searching in other quadrants or even start scrabbling at the pool wall. The latter behaviour we have attributed to anticipation of the subjects of being removed from the pool by the experimenter (Devan and McDonald, 2001), a different source of reinforcement in the MWM task.

For a 60 second trial duration, in our experience this is plenty of time for the mice to find the platform particularly after the first few trials have occurred and they start finding the platform. Similarly, if longer durations are used the mice can become cold and this can impact their performance on the task.

The reviewer also raises concerns about swim speed as a measure of motor function. Swim speed, we think is a legitimate measure of motor function. It is very difficult to swim at the same speed as a normal subject if motor control is significantly compromised. As our data shows, the different groups show similar swim speed (between groups) during MWM training and there is no change in swim speed throughout training (within or between groups).

We do agree with reviewer that determining search strategies can be more sophisticated than only swim speed and we have used these approaches recently (McDonald and Hong, 2020). For future work we will employ these approaches.

Additionally, we replaced the MWT with standard term MWM in text as well as in Fig. 1 in the revised manuscript.

We thank the reviewer for making useful suggestions on the manuscript. Your suggestions improved the manuscript significantly.

References

- Devan B D and McDonald R J. A cautionary note on interpreting the effects of partial reinforcement on place learning performance in the water maze. *Behav Brain Res.* 2001 Mar 15;119(2):213-6.
- Mehla et al. Looking beyond the standard version of the Morris water task in the assessment of mouse models of cognitive deficits. *Hippocampus.* 2019 Jan;29(1):3-14.
- McDonald, R.J., and Hong, N.S. (2020) The complex path of behavioural analysis: the past, present, and future. *F1000*, Jan 21.

Reviewer #2 (Remarks to the Author):

In this article, the authors investigated the effects of cognitive training for memory impairments and brain pathology in Alzheimer's disease (AD) using a knock-in mouse model. They found that multi-domain cognitive training prevented more effectively cognitive deficits in AppNL-G-F mice in several behavioral tests. In addition, A β deposition and microgliosis were ameliorated in memory-related brain regions, and cholinergic neuron positive for ChAT were more preserved in

AppNL-G-F mice with multi-domain cognitive training. Their results may provide evidence that cognitive training or cognitive enhancement have beneficial effects to prevent cognitive decline in AD patients. The study is well designed and conducted, though the reviewer raises several suggestions below to revise the manuscript for publication.

Comment 1: In the Morris water task, AppNL-G-F mice with single- or multi-domain cognitive training exhibit less thigmotaxic behavior compared with the mice with no training. As the authors described, thigmotaxic behavior is shown to reflect animal's emotional state, whereas it may be associated with alterations of search strategy to reach the platform. Can the authors discuss these points further? Since some reports says that AppNL-G-F mice exhibit more anxiolytic behaviors in other tests (e.g. open field, elevated plus maze), for the author's conclusion, this reviewer thinks that it is better to evaluate anxiety-like behavior in AppNL-G-F mice with cognitive training using other tests.

Reply: We thank the reviewer for their critical review of the manuscript. We agree with the reviewer on this point. We have expanded that section in the manuscript and clearly state our position that current evidence suggests that this mouse model has an anxiety phenotype and provide some citations:

The following sub-section has been added to the revised manuscript on page 20-21 under the heading "Discussion".

"APP mice, anxiety, and cognitive training"

It has been reported that the APP NL-G-F mouse exhibits an anxiety phenotype (Pervolaraki et al., 2019; Locci et al., 2021) although there is no consensus on this issue (Sakakibara et al., 2018; Maezono et al., 2020; Kundu et al., 2021; Emre et al., 2022). The results of the present study suggest that this knock-in mouse model of AD might exhibit anxiety and other reports suggest there is an anxiety phenotype using other measures including elevated plus maze, light/dark test, and open field that emerge quite early (Auta et al., 2022; Sakakibara et al., 2018).

It is important to account for this phenotype because anxiety could impair performance on tasks like the MWM independent of learning and memory function and simply reducing anxiety via a treatment like cognitive training could improve performance.

Our measure of anxiety in the present study was thigmotaxia (not swimming away from the pool wall) and it has been shown that this behaviour can impair MWM performance (Huang et al., 2012; Küçük et al., 2008). It is possible that the multi-domain cognitive training improved MWM performance simply by reducing anxiety.

However, as we have argued recently (Mehla et al., 2022) thigmotaxia can emerge for a variety of reasons including altered plasticity and associated learning and memory impairments, anxiety, executive functions, and gonadal hormones. It is unclear in studies of this type what is driving the increase in thigmotaxia in the APP mice and how cognitive training reverses these impairments.

We envisioned at least three potential explanations as to what is driving the impairments on the MWM in the APP mice. First, the AD pathology in the APP mice impairs plasticity producing associated learning and memory impairments in the HPC and related neural circuits that render the subjects less confident in finding the escape platform and so they remain near the pool wall. Second, the AD pathology in the APP mice impairs neural networks important for controlling anxiety that are separate from those important for learning and memory functions centered on the HPC neural network.

Finally, the hypothesis that we think is most likely is that a combination of learning and memory dysfunction and increased anxiety is driving the impairments in MWM performance in the APP mice. Viewed through this lens, the improved performance on the MWM in the APP mice following multi-domain cognitive training is via both mechanisms, but how?

One hint comes from an interesting pattern emerging from decades of research on the neural basis of learning and memory and parallel work investigating brain systems implicated in anxiety. This work implicates similar neural systems in both learning and memory functions and in controlling fear responses and general anxiety (Bannerman et al., 2004; McHugh et al., 2004; Trow et al., 2017). Key brain regions that have been identified include HPC, amygdala, and various parts of the prefrontal cortex.

These regions have been implicated in reducing fear and anxiety via learning and memory functions by constraining fear responses to predictive contexts/cues predictive of aversive events (McDonald et al., 2017). Thus, it is likely that the impacts of multi-domain cognitive training on anxiety are via reduced pathology and dysfunction of the HPC, amygdala, and prefrontal cortex and associated cognitive functions. This analysis is consistent with the claim that voluntary exercise improved learning and memory functions at least partially via reductions in anxiety and reduced various brain pathologies associated with AD in medial temporal lobe brain regions thought to be central in complex neural networks supporting various forms of memory that control fear and anxiety responses.

Our future work using these mouse models of AD will focus on potential impacts of AD pathology on dorsal and ventral HPC as the latter has been implicated in anxiety and the former in precise spatial navigation. We are also interested in assessing the impacts on orbital frontal cortex as we have found this region to be specifically involved in generalized anxiety but not the learning and memory processes involved in context fear conditioning [Zelinski et al., 2010; Trow et al., 2016].”

Reference

- Auta, J., Locci, A., Guidotti, A., Davis, J. M. & Dong, H. Sex-dependent sensitivity to positive allosteric modulation of GABA action in an APP knock-in mouse model of Alzheimer’s disease: Potential epigenetic regulation. *Curr. Res. Neurobiol.* 3, 100025 (2022).
- Bannerman DM, Rawlins JN, McHugh SB, Deacon RM, Yee BK, Bast T, et al. Regional dissociations within the hippocampus--memory and anxiety. *Neurosci Biobehav Rev.* 2004;28:273.

- Emre C, Arroyo-García LE, Do KV, Jun B, Ohshima M, Alcalde SG, et al. Intranasal delivery of pro-resolving lipid mediators rescues memory and gamma oscillation impairment in App NL-G-F/NL-G-F mice. *Commun Biol.* 2022;5:245.
- Huang Y, Zhou W, Zhang Y. Bright lighting conditions during testing increase thigmotaxis and impair water maze performance in BALB/c mice. *Behav Brain Res.* 2012;226:26.
- Küçük A, Gölgeci A, Saraymen R, Koç N. Effects of age and anxiety on learning and memory. *Behav Brain Res.* 2008;195:147.
- Kundu P, Torres ERS, Stagaman K, Kasschau K, Okhovat M, Holden S, et al. Integrated analysis of behavioral, epigenetic, and gut microbiome analyses in App NL-G-F, App NL-F, and wild type mice. *Sci Rep.* 2021;11:4678.
- Locci A, Orellana H, Rodriguez G, Gottliebson M, McClarty B, Dominguez S, et al. Comparison of memory, affective behavior, and neuropathology in APPNLGF knock-in mice to 5xFAD and APP/PS1 mice. *Behav Brain Res.* 2021;404:113192.
- Maezono SEB, Kanuka M, Tatsuzawa C, Morita M, Kawano T, Kashiwagi M, et al. Progressive Changes in Sleep and Its Relations to Amyloid- β Distribution and Learning in Single App Knock-In Mice. *eNeuro.* 2020;7:ENEURO.0093-20.2020.
- McDonald RJ, Hong NS, Devan BD. Interactions among multiple parallel learning and memory systems in the mammalian brain. Chapter published in Howard Eichenbaum (Ed). *Learning and Memory: A Comprehensive Reference (2nd Edition)*. 2017.
- McHugh SB, Deacon RM, Rawlins JN, Bannerman DM. Amygdala and ventral hippocampus contribute differentially to mechanisms of fear and anxiety. *Behav Neurosci.* 2004;118:63.
- Mehla et al. Dramatic impacts on brain pathology, anxiety, and cognitive function in the knock-in APPNL-G-F mouse model of Alzheimer disease following long-term voluntary exercise. *Alzheimers Res Ther.* 2022 Sep 30;14(1):143.
- Pervolaraki E, Hall SP, Foresteire D, Saito T, Saido TC, Whittington MA, et al. Insoluble A β overexpression in an App knock-in mouse model alters microstructure and gamma oscillations in the prefrontal cortex, affecting anxiety-related behaviours. *Dis Model Mech.* 2019;12:dmm040550.
- Sakakibara Y, Sekiya M, Saito T, Saido TC, Iijima KM. Cognitive and emotional alterations in App knock-in mouse models of A β amyloidosis. *BMC Neurosci.* 2018;19:46.
- Trow, J. E. et al. Evidence of a role for orbital prefrontal cortex in preventing over-generalization to moderate predictors of biologically significant events. *Neuroscience* 345, 49–63 (2017).
- Zelinski, E. L., Hong, N. S., Tyndall, A. V, Halsall, B. & McDonald, R. J. Prefrontal cortical contributions during discriminative fear conditioning, extinction, and spontaneous recovery in rats. *Exp. brain Res.* 203, 285–97 (2010).

Comment 2: As the authors mentioned, multi-domain cognitive training also improved the performance of 12-month-old AppNL-G-F mice in the novel object recognition test. The mice exposed to single-domain training showed low levels of exploring objects compared to other 3 groups. This reviewer thinks that low exploratory activity may result in lower discrimination index. Did these mice show similar levels of exploration during the habituation?

Reply: We were concerned with this issue as well and found that low exploratory activity was also observed during the habituation/ or training phase. (this data was provided in the supplementary figures previously) We have provided this data in supplementary figure 2 A.

Comment 3: Histological analyses revealed that multi-domain cognitive training ameliorate A β deposition in AppNL-G-F mice, however, there were some regional differences for the phenotype. As the data presented, multi-domain cognitive training can ameliorate A β deposition in some regions such as the hippocampus and perirhinal cortex, but not in the medial prefrontal cortex. Can the authors discuss for the regional differences that multi-domain cognitive training affect the brain A β pathology? Indeed, decreased number of small-sized A β plaques in some regions can raise the possibility that multi-domain training suppress A β production or accelerate its clearance. Can the authors discuss how cognitive training affect A β metabolism? As a minor point related with the data, this reviewer recommends replacement of the image showing A β deposition at section (-3.08mm) in AppNL-G-F mice with single-domain cognitive training.

Reply: This is an interesting phenomenon (regional specific effects of MT on A β). One idea we would proffer is that the tasks selected would not activate the MPFC as much as it would the three different learning and memory networks in the medial temporal lobes. We have added this idea in the discussion on page 21-22.

Your idea about small-sized A β plaques was intriguing and we have mentioned this as a potential mechanism that deserves attention in future work. We have added a few sentences on this issue in the discussion on page 23.

Although not a lot is known about the mechanisms mediating improved cognitive outcomes in AD following cognitive training there is some work suggesting multiple targets. We have added a paragraph in the discussion exploring these ideas.

We have also replaced that image as requested.

“Prior research in aging humans suggests that improved learning and memory functions following multidomain training is complex and probably targets multiple mechanisms including improving synaptic plasticity, increase blood flow in the brain, improve beta-amyloid clearance, and activate more of the brain. Some recent human AD studies show that cognitive training can improve cognition and attentional process and these changes correlate with changes in the ratios of APP isoforms (Casoli et al., 2020). Furthermore, one potential mechanism for the region-specific reduction of A β plaques could be via a reduction of the number of small-sized A β plaques in some regions can raise the possibility that multi-domain training suppresses A β

production or accelerate its clearance. Future work is required to assess this intriguing possibility.”

Reference

- Casoli, T., Giuli, C., Balietti, M., Fabbietti, P. & Conti, F. Effect of a Cognitive Training Program on the Platelet APP Ratio in Patients with Alzheimer’s Disease. *Int. J. Mol. Sci.* 21, (2020).

Comment 4: AppNL-G-F mice with single- and multi-domain training exhibit reduced microgliosis in the medial prefrontal cortex as shown by Iba1 staining. It is curious because these mice have similar levels of A β deposition in this region compared with the mice with no training. Can the authors further discuss these points?

We agree with the reviewer that this is an interesting pattern of effects. We have added a short paragraph in the discussion (page 21-22) about this issue.

“This points to the possibility that cognitive training can act on multiple mechanisms and via different mechanisms in different brain regions. For example, in the medial temporal lobe multi-domain cognitive training might reduce A β depositions and inflammation emanating from microglial sources. The mPFC might be more sensitive to both ST and MT training which can have positive effects on microglial inflammation but does not reduce A β . Further research is required to understand the effects of different forms of cognitive training on different sources of inflammation and any regional-specific effects. We also need to explore what the functional impacts of reduced inflammation in mPFC following cognitive training has on tasks that tap into that frontal system.”

Also, this reviewer wonders if multi-domain cognitive training ameliorates astrogliosis as well. Since astrocyte has known to be involved in various homeostatic functions including neuronal support, the data would be of importance to speculate the underlying mechanisms that cognitive training can ameliorate behavioral deficits or brain pathology.

Reply: We agree that astrogliosis might also have been impacted but we did not measure this source of inflammation. Our future work will assess both sources of inflammation. We thank the reviewer for this suggestion.

Comment 5: Interestingly, AppNL-G-F mice exposed to multi-domain cognitive training exhibited very few A β plaques in the MSDB complex than other brain regions. And, they also preserved the relative number of ChAT-positive cholinergic neurons. Do the authors have any data to assess brain acetylcholine content or cholinergic projections to support that cognitive stimulation enhance forebrain cholinergic function? And, can the authors discuss underlying

mechanisms that cognitive training preserve cholinergic system? Previous studies show that NGF-p75NTR signaling has a critical role for maintaining cholinergic neuron survival.

Reply: We appreciate the reviewer's useful comment, and we totally agree with their suggestion that it would be interesting to comment on the brain acetylcholine (ACh) content or cholinergic projections to support the finding that cognitive stimulation enhanced forebrain cholinergic function. Unfortunately, we have not measured ACh levels and cholinergic projections in the brains of the mice in the current study. Moreover, we could not find any studies where the effects of cognitive training on the ACh production/ or levels and cholinergic projections has been reported. Additionally, we did not find evidence in the research literature implicating NGF-p75NTR signaling in the protective effects of cognitive training in AD mice. We thank the reviewer for these great suggestions. We are currently completing work assessing the effects of multi-domain training on ACh output in the hippocampus using acetylcholinesterase staining techniques. We were not planning on assessing cholinergic fiber density and NGF-p75NTR signaling but based on the reviewers comments we will incorporate these fascinating potential mechanisms as well. We thank the reviewer for these suggestions particularly the NGF idea. We have added this future direction in the discussion on page 22.

Comment 6: As the authors described, 3 behavioral tests (Morris water task, novel object recognition test, and fear conditioning test) were performed for cognitive training and tests. Some behavioral tests are established for assessing cognitive function and memory in mice, so are there any reasons why the authors included these 3 tests?

Reply: We appreciate the reviewer's useful comment. The other reviewer raised the same issue and we have explained the logic in that response above and in the text of the manuscript now (found in the introduction).

Comment 7: What order did the authors conduct the behavioral tests in multi-domain cognitive training? Some behavioral tests are stressful to mice. In these 3 tests It may be the order that novel object recognition, Morris water task, and fear conditioning to arrange from less to more stressful in these 3 tests. If possible, the information may be included in Figure 1.

Reply: We totally agree with the reviewer's concern about the order of behavioral tests. We used the following order for behavioral tests in the present study and we have incorporated this important information in the Figure 1 as well as in Multi-domain training group section page 6 in the revised manuscript:

We used the following order for the behavioral tests: novel object recognition, MWM, fear conditioning to expose the mice in a sequence from less to more stress in these neurobehavioral tests at the beginning of "Multi-domain training group experiment" (page 6).

Comment 8: It may be more helpful for readers in other fields to include the information for animal numbers used in each behavioral and histological analysis in each Figure legends.

Reply: As per the reviewer's suggestion, we have now included the animal number in the figure legends. The changes are highlighted in green color.

We thank the reviewers as well for the quality of comments they provided. Their suggestions were fair and improved the manuscript.

REVIEWERS' COMMENTS:

Reviewer #1 (Remarks to the Author):

Criticisms have been addressed

Reviewer #2 (Remarks to the Author):

The authors included some insightful discussions to improve their manuscript. Moreover, they added important information for their experimental design. This reviewer has no more suggestions for revise and is looking forward to their future works.